



# Double moment normalization of hail size number distributions over Switzerland

Alfonso Ferrone[1, 2], Jérôme Kopp[3], Martin Lainer[2], Marco Gabella[2], Urs Germann[2], and Alexis Berne[1]

[1]Environmental Remote Sensing Laboratory, École Polytechnique Fédérale de Lausanne (EPFL), Lausanne, Switzerland
[2]Federal Office of Meteorology and Climatology MeteoSwiss, Locarno-Monti, Switzerland
[3]Oeschger Centre for Climate Change Research and Institute of Geography, University of Bern, Bern, Switzerland

**Correspondence:** Alexis Berne (alexis.berne@epfl.ch)

**Abstract.** Measurements of hailstone diameters and kinetic energy, collected by the Swiss network of automatic hail sensors, are available in three regions of Switzerland for the period between September 2018 and August 2023. In this study, we propose the use of double moment normalization for modeling the hail size number distribution (HSND), which is defined as the number of hailstone impacts measured, for each diameter size, by one instrument during one hail event. This method uses two of the empirical moments of the HSND to compute a normalized distribution. While the HSND is dependent on the duration and intensity of the event and on the detection area of the sensor, we show that the normalized distribution has limited variability across the three geographical regions of deployment of the sensors. Thanks to its invariance in space and time, a generalized gamma is used to model the normalized distribution, and its parameters have been determined through a fit over approximately 70% of the events. The fitted model and the previously chosen pair of empirical moments can be used to reconstruct the HSND at any location in Switzerland. The accuracy of the reconstruction has been estimated over the remaining 30% of the dataset. An additional evaluation has been performed on an independent HSND, made of estimates of hail diameters measured by drone photogrammetry during a single event. This HSND has a much larger number of hailstone impacts (18000) than those of the hail sensor events (from 30 to 400). The double moment normalization is able to reproduce well the HSND recorded by the hail sensors and the drone, albeit with an underestimation of the number of impacts at small diameters. These results highlight the invariance of the normalized distribution and the adaptability of the method to different data sources.

## 1 Introduction

Hail has been shown to be the cause of severe damage to properties and crops in Europe (Púčik et al., 2019), the USA (Brown et al., 2015), and Australia (Warren et al., 2020), and it has therefore been an active topic of research for several decades. In Switzerland, the focus of the current study, there is currently a large interest in improving the monitoring, nowcasting, and climatological description of this weather phenomenon.

Work has been conducted on studying the spatial and temporal distribution of hailstorms (Nisi et al., 2016), producing a climatology of hail streaks over the Alpine region (Nisi et al., 2018), and investigating the use of crowdsourced hail reports (Barras et al., 2019). A collaboration between the Federal Office of Meteorology and Climatology (MeteoSwiss) and partners from various sectors in the Hail Climate Switzerland project (NCCS, 2021) resulted in a uniform national reference on hail





hazards. Finally, the ongoing scClim project (ETH Zurich, 2023), involving a series of public and private partners, aims to establish a seamless model chain from thunderstorm simulations to the quantification of hail impacts.

In 2015, MeteoSwiss installed a small pilot network with seven stations in the Napf region to test a new automatic hail sensor of inNET Monitoring AG (Wetzel, 2018). In June 2018, within a collaboration between MeteoSwiss, "La Mobilière", inNET Monitoring AG and the University of Bern, work began on the installation of a wider network of 80 hail sensors. The
instruments in this network operate continuously and autonomously, converting the vibration caused by the impact of individual hailstones into estimates of kinetic energy and diameter. The resulting dataset, which covers multiple years of operation, offers the opportunity for a variety of studies on the hail size. For instance, Kopp et al. (2023b) focus on hail events over a single season, while Kopp et al. (2023a) provide an analysis of the distribution of all hail diameters recorded since 2018. Using a similar multi-year dataset, expanded by the measurement collected until 1 September 2023, the current article proposes a way
to model the hail size number distribution (HSND) for individual hail events. The HSND has been defined as the number of impacts recorded by one instrument for each diameter value, and it is dependent on the detection area of the instrument and the duration of the event. It should be noted that this quantity differs from the hail size distribution (HSD) investigated in a large fraction of the past scientific literature, since the HSD is usually computed over a unit area and for a fixed duration of time.

While the existence of automatic hail sensors is a new development in the study of distributions of hail sizes, research on the
HSD has been conducted by the scientific community for decades using a variety of instruments (Changnon, 1969). Among them, hailpads resemble in some aspects the hail sensors, they both measure hail at ground level and have a surface of similar scale. While measurement campaigns using these instruments have taken place over Switzerland in past years (Federer et al., 1986), no contemporary dataset is available in the region of deployment of the hail sensors. However, measurements from hailpads collected in northern Italy have been used by Kopp et al. (2023a) to perform a comparison between the two types of
instruments. The study found that their distribution of diameter sizes, aggregated over multiple hail events, are almost identical for diameters up to 18 mm.

Spectrometers equipped with a hail/rain separator (Federer and Waldvogel, 1975) and "hail catchers" (Cheng and English, 1983) have also been used in early studies on hail size. Some of these data sources provided the basis for the development of the exponential model of the hail size distribution, originally formulated and expanded upon in the decades following the
1960s (Douglas, 1963; Waldvogel et al., 1978). As mentioned previously, the object of these analyses differs from the HSND, since it is normalized in both area and time. This quantity is typically indicated by $N(D)$, with $D$ denoting the diameter and the quantity $N(D)dD$ providing the number of hailstones whose size is between $D$ and $D + dD$. The exponential model for the hail size distribution is usually formulated as follows:

$$N(D) = N_0 e^{-\Lambda D}.$$

In the expression above, $N_0$ (same units as $N(D)$) and $\Lambda$ (inverse units of $D$) are two of the parameters of the exponential model, while a third one ($D_{max}$, using the same units as $D$) is provided by the condition $5\,\text{mm} < D < D_{max}$. The relationship between these parameters, possibly leading to a simplified model with only two independent parameters, as well as their link to other physical quantities, such as the kinetic energy and mass fluxes, has been explored by Ulbrich and Atlas (1982).





While the exponential relationship constitutes a valid representation of the HSD, the relatively large size of the dataset collected by the hail sensors allows us to experiment with an alternative approach. In this study, we propose the use of double moment normalization to model the HSND, following the work conducted by Lee et al. (2004) for the drop size distribution (DSD). Similarly to the hail case, an exponential relationship between $N(D)$ and $D$ to represent the DSD has been proposed by early studies on the topic (Marshall and Palmer, 1948), albeit with some differences, such as the absence of a minimum and maximum value for $D$. Later studies have proposed a generalization, of which the exponential model represents a particular case, in the form of single moment normalization (Sempere-Torres et al., 1994, 1998). To address some issues with different rainfall types, Lee et al. (2004) introduced the double moment normalization, which allows representing a DSD using two of its moments and the knowledge of the "normalized distribution". The latter provides the overall "shape" of the DSD and is assumed to remain the same over the geographical region and period of time for which it is used.

A clear benefit of using the double moment normalization, at least in the case of liquid precipitation, lies in the links that can be found between the empirical moments of the distribution and some radar variables. This relationship allows the retrieval of the DSD from radar measurements, covering areas much larger than the one that would be possible to cover by in-situ instruments, such as disdrometers. This option has been explored by Raupach and Berne (2017) in the case of rain, it has been expanded to the small drop sizes of the DSD (called the drizzle mode) by Raupach et al. (2019), and examples of research on the topic can be found in contemporary scientific literature (Lee et al., 2023). Using a similar model for the HSND would, in theory, allow future studies to mimic the aforementioned works, ultimately allowing for the retrieval of the HSND from radar scans. The applicability of the double moment normalization to hail measurements has already been demonstrated by Field et al. (2019), using data collected by an airplane inside thunderstorm clouds. This measurement setup differs from the one used in the current article since the hail sensor network provides diameter estimates at the ground. Phenomena occurring on the falling hailstones, such as melting (Mason, 1956), may affect their size, leading to some differences in the HSD. Other differences between the current analysis and the one presented by Field et al. (2019) include the evaluation of the spatial invariability of the normalized distribution and the usage of different moment orders.

The usage of a storm-penetrating aircraft does not represent the sole possible source of airborne measurements of hail size. The study conducted by Lainer et al. (2023) uses images collected by a drone, over a relatively flat and uniform surface, to retrieve the projection on the view plane at the ground of the major and minor axis of hailstones. In their study, the distribution of these estimates of hail sizes is compared to the ones captured by 3 neighboring hail sensors. Thanks to the relatively large surface covered by the drone ($750 \ \mathrm{m}^2$), the number of hailstones recorded during a single event greatly exceeds the typical number of impacts on an individual hail sensor. The same dataset of drone-derived hail size estimates is used in the current study to evaluate how the parameters of the double moment normalization, retrieved solely from hail sensor data, can be applied to HSND retrieved from a different data source.

This comparison with the drone-derived data also allows us to evaluate the validity of using the HSND rather than the HSD for our analysis. As mentioned previously, the quantity that we model is not normalized over a unit surface and a fixed time duration, and throughout the study, it will be denoted by $N_u(D)$ (where the subscript $u$ stands for "unnormalized"). While $N(D)$ allows for a more direct comparison between different data sources, $N_u(D)$ offers an advantage in the simplicity





of its computation, which requires no knowledge of the precise area of detection, timing of the impacts, or duration of the

event. These last two aspects become relevant for distributions derived a posteriori, as is the case for the drone. However, this lack of normalization leads to considerable differences in the values of $N_u(D)$, which become particularly noticeable in the comparison between drone and hail sensor. Since the parameters of the double moment normalization have been computed solely on the hail sensor data, the performances of the method on the drone HSND will provide information on its independence from the event duration and the area over which measurements have been collected.

The current study is structured as follows. Section 2 introduces the datasets and their processing, leading to the definition of the HSNDs. Section 3 describes the method, including the theory behind the double moment normalization and the error metrics used throughout the analysis. The results pertaining to the normalized distributions are presented in section 5, while the ones relative to the HSNDs are in section 6. Finally, section 7 provides a summary and the conclusions of the study.

## 2   Data

In this section, we present the two datasets used in the current analysis. Both datasets are comprised of measurements of hail diameters: the first one is a set of HSNDs collected by the Swiss network of automatic hail sensors, and the second one is a single distribution of hail sizes recorded through drone photogrammetry.

### 2.1   Hail sensors

In the period between 2018 and 2020, 80 automatic hail sensors have been deployed in three regions, representing the three

main hot spots for hail in Switzerland: Jura (15 sensors) and Napf (38 sensors) north of the Alps, and Ticino (27 sensors) south of them. Among the three regions, Ticino stands out as the one where the largest hailstones have been recorded. Their locations are shown in Figure 1. While the microphysical mechanisms involved in the formation of hail are expected to be the same everywhere, it has been found that hailstorm frequency and intensity differ between the three regions (Feldmann et al., 2023). Additional information on the relative positions of the instruments in each region is visible in the three panels at the bottom of

the figure. While detailed information on the hail sensors and their measurements has been presented by Kopp et al. (2023a), here we summarize a series of relevant information for the current analysis.

The hail sensor converts the oscillations caused by the impact of falling hailstones on a Makrolon disc of approximately $0.2 \ \mathrm{m}^2$ into estimates of kinetic energy. The latter is, in turn, converted into an estimate of the diameter ($D$) of the falling hailstone. The conversion is performed by assuming that the hailstone is spherical and has a constant drag coefficient (Löffler-

Mang et al., 2011).

The dataset used in this study spans the period between 9 September 2018 and 1 September 2023. We use the same radar reflectivity filter (> 35 dBZ) as in (Kopp et al., 2023a) to ensure that there is a storm environment in close vicinity of the sensor and minimize the risk of impacts not due to hailstones. We end up with 15902 hailstone impacts.



### 2.1.1 Minimum hailstone size

In addition to the previously mentioned conditions, we applied a minimum threshold on the size of hailstones, set to $D > 5\,\mathrm{mm}$, following the definition provided by the Glossary of Meteorology (American Meteorological Society, 2023). This condition reduces the dataset size to 13926 impacts. As described by Kopp et al. (2023a), in the processing of the raw measurements of the sensors, a threshold controls the minimum detectable hail size, and diameters below this threshold are excluded. Until 2022, the threshold was not adjusted every time a sensor was re-calibrated, resulting in a varying lower limit of hail size diameters.

As of 2023, the calibration procedure has been changed and the lower threshold is systematically set back to $5\,\mathrm{mm}$.

This variation in the lower limit led to the recording of diameters below $5\,\mathrm{mm}$. The condition $D > 5\,\mathrm{mm}$ ensures that such impacts are excluded from our study. HSNDs truncated at a minimum diameter higher than $5\,\mathrm{mm}$ also happened in some cases. While the minimum diameter can reach up to $8\,\mathrm{mm}$ in a handful of extreme cases, minimum values of $6\,\mathrm{mm}$ or even $7\,\mathrm{mm}$ are relatively more common. Excluding all measurements below these thresholds would, however, greatly reduce the number

of data available for the analysis, undermining its robustness, which relies on the availability of a dataset as large as possible in some steps (e.g. fit of the generalized gamma function over the normalized distributions, described in section 3.1). Therefore, rather than increasing the minimum threshold, we decided to estimate the impact of the missing diameters in the lower end of the distribution on the value of its empirical moments, which play a crucial role in the double moment normalization of the HSNDs. This analysis is presented in section 4.

### 2.1.2 Definition of the hail events

Once the dataset has been processed, we analyze the time series of impacts recorded by each hail sensor to identify separate events. Each of them has been defined as a period in which hailstones are recorded with a gap between consecutive impacts of less than 20 minutes, corresponding to the largest blank period studied in Kopp et al. (2023a).

Our implementation of the double moment normalization follows similar studies, in which the method has been applied to

rain (Raupach and Berne, 2017) and drizzle (Raupach et al., 2019) measurements. These articles, however, utilize datasets considerably larger than the one recorded by the Swiss hail sensor network, as can be expected given the relatively low frequency of occurrence of hail compared to rain. Therefore, the method needs to be adapted to accommodate the lower number of measurements available. We decided to use the "hail event" as the smallest temporal unit over which the HSND is computed, without further subdividing it into intervals with a fixed time duration.

The number of hailstones recorded in the available events varies considerably. In some cases, the number can be too low for a meaningful computation of the empirical moments. Therefore, a threshold on the minimum number of impacts for each event has to be enforced. We performed a series of tests to estimate its effect on the dataset size, by setting this minimum number to the following values: 20, 25, 30, 35, 40, 45, 50. The number of events in the dataset varies between 125 (threshold equal to 20) and 65 (threshold equal to 50).

A compromise value equal to 30 impacts has been chosen. The total number of hailstones in the events that satisfy this threshold is 9113. While there is a degree of arbitrariness behind this choice, the threshold value has been selected for the





relatively large size of the resulting dataset (94 events) while also having a limited impact on the largest diameters available in the various HSNDs. In particular, we focused on impacts for which $D > 15$ mm, which are rare and particularly valuable, for their importance in accurately representing the tail of the HSND. For these relatively large diameters, imposing a minimum

threshold equal to 30 removes only 2 more hailstones than the smallest threshold (20). This number increases for larger minimum values, as more events are excluded, reaching 9 additional impacts removed for the highest threshold (50).

The resulting HSNDs for all events are displayed as a 2-dimensional histogram in Figure 2. The binning of $D$ has been performed using a constant width $dD = 1$ mm, and the number of impacts recorded at each bin is $N_u(D)dD$. The figure shows a relatively large difference in the value of $N_u(D)$ among the events, especially for the smallest diameters. Another noticeable

aspect of Figure 2 is the scarcity of measurements for the largest diameters (e.g. above 20 mm), for which a majority of the events do not have available data. The potential impact of the missing measurements at the tail of the HSND is further explored in Section 4, in which we address their effect on the value of the empirical moments.

## 2.2 Drone observations

Estimates of hail size derived from drone photogrammetry (Lainer et al., 2023) constitute the second data source in this

study. The distribution of diameters derived from the drone observations during a hail event near Entlebuch (Canton Lucerne, Switzerland) on 20 June 2021, over a field with an area of approximately 750 m$^2$, is used for validation purposes. Their independent origin allows us to verify whether the double moment normalization derived from the hail sensor measurements captures the intrinsic qualities of the HSND. Additionally, thanks to the larger values of $N_u(D)$ recorded by the drone, we can test how well the results scale to different sizes of the HSND.

The starting points of our analysis are the minimum and maximum diameter of hailstones, projected onto the image plane as seen by the drone, derived by Lainer et al. (2023). Using each pair of values as the axes of an ellipse, we can compute the diameter of a circle with the same area. This value, referred to as the equivalent diameter, is used in the rest of the analysis.

To allow a meaningful comparison with the measurements collected by the hail sensor, we decided to impose the minimum threshold $D > 5$ mm on the drone HSND. The value of $N_u(D)$ for each discrete diameter bin (2 mm intervals) is shown as

orange circles in Figure 2.

Figure 2 also provides us with a direct comparison between the HSNDs recorded by the drone and the hail sensors. Due to the difference in measuring area (approximately 750 m$^2$ for the drone and 0.2 m$^2$ for each hail sensor), the drone stands out for the much larger values of $N_u(D)$, which can reach up to several thousands of impacts for the smallest diameters. At higher values of $D$, the availability of a relatively high number of hailstones allows us to evaluate the double moment normalization

on diameters rarely observed in the hail sensor dataset.

## 3 Method

This section provides a brief introduction to the theory behind the double moment normalization and its implementation, as well as the definition of a series of error metrics used throughout the analysis.





### 3.1 Double moment normalization

As mentioned in the introduction, the double moment normalization was originally introduced by Lee et al. (2004), with the explicit aim of providing a general representation of drop size distributions. However, the structure of our analysis follows more closely the one performed by Raupach et al. (2019), and we will therefore borrow the notation of the latter throughout our study. Please note that the underlying theory between the aforementioned studies is the same and, in the current section, it has been adapted to the use on HSNDs.

The starting point of our analysis is the hailstone diameters ($D$) available from the hail sensor and drone datasets. In the previous section, we introduced the quantity $N_u(D)$, representing the number of impacts recorded at a certain value of $D$. Visual representations of the HSND, such as the one in Figure 2, require a discrete binning of the range of diameter values, in order to compute the value $N_u(D)$. The choice of the size of these bins is arbitrary, and we assume that, in theory, hailstone sizes vary continuously. Under this continuous assumption, we can better redefine $N_u(D)dD$ as the number of impacts recorded in an interval $D + dD$, where $dD$ has an infinitesimal size. We can therefore define the moment of order $p$ of the distributions as:

$$M_p = \int_0^\infty N_u(D)D^p dD. \tag{1}$$

In practice, for an event with a finite number of impacts, the integral becomes a sum over all the measured diameters, elevated to a power equal to $p$.

Given two different orders, $i$ and $j$, we can use the corresponding empirical moments $M_i$ and $M_j$ to define a unitless quantity called "normalized diameter", indicated by the symbol $x$, and computed using the following formula:

$$x = \frac{M_i^{1/(j-i)}}{M_j^{1/(j-i)}}D \tag{2}$$

This quantity allows us to compute the "normalized distribution", denoted by $h(x)$, and linked to the empirical moments and to the original HSND by the equation:

$$h(x) = \frac{M_j^{(i+1)/(j-i)}}{M_i^{(j+1)/(j-i)}}N_u(D) \tag{3}$$

By using HSNDs as input, rather than HSDs, we hypothesize that the normalization behind the computation of $h(x)$ allows the direct comparison of normalized distributions from events of different durations and recorded over different areas. In practical terms, once a bin size is defined for $x$, we can compute the value of $h(x)$ for each event in our dataset. This provides us with a set of normalized distributions based on hail measurements, which can be used to estimate an analytical counterpart. This counterpart, denoted by $\hat{h}(x)$, is obtained by fitting a generalized gamma distribution over $h(x)$. In our case, the fit is performed over a subset of all available events: the training set, defined in section 5.1. Our approach differs from the one described by Raupach et al. (2019) since, in our case, the fit uses the values of $h(x)$ from all events, instead of the median or average values at each discrete $x$ bin.





The analytical form chosen for $\hat{h}(x)$ is:

$$\hat{h}(x) = c \frac{\Gamma_i^{(j+c\mu)/(i-j)}}{\Gamma_j^{(j+c\mu)/(i-j)}} x^{c\mu-1} \exp\left[ -\left( \frac{\Gamma_i}{\Gamma_j} \right)^{c/(i-j)} x^c \right], \tag{4}$$

where $\mu$ and $c$ are the parameters of the generalized gamma distribution, $\Gamma$ represents the gamma function, and the quantities $\Gamma_p$ (with $p=i$ or $p=j$) are defined by $\Gamma_p = \Gamma(\mu + p/c)$.

As written, equation 4 involves ratios of large numbers, resulting from elevating $\Gamma_i$ and $\Gamma_j$ to an exponent depending on the moment orders and the parameters $\mu$ and $c$. During the fitting procedure, as the algorithm modifies the value of $\mu$ and $c$, issues can arise when any factor in equation 4 exceeds the maximum value allowed for the variable that holds the result. Even when
this limit is not exceeded, performing a ratio between very large numbers can result in a loss of precision. Therefore, we decided to replace the $\Gamma$ functions with its logarithm, using the "loggamma" function from the Python library "scipy" (Virtanen et al., 2020). This allows us, in turn, to perform subtractions and multiplications in place of the original ratios and exponentiation. Once these operations have been performed, we compute the exponential of the results, giving us the same value that we would originally have obtained from equation 4. This procedure effectively increases the range of suitable values for $\mu$ and $c$, which
has been set to $[10^{-6}, 500]$ for both parameters during the fitting procedure.

After implementing these adjustments to $\hat{h}(x)$, the fit has been performed by minimizing the root mean square error (RMSE, defined in section 3.2) between the logarithm of the various $h(x)$ in the training set and the logarithm of $\hat{h}(x)$. The parameters $\mu$ and $c$ have both been initialized with the value 1, which would result in $\hat{h}(x)$ having a simple exponential form. Once $\hat{h}(x)$ has been estimated, it is used alongside the empirical moments $M_i$ and $M_j$ to compute an estimated HSND, indicated by
$\hat{N}_u(D)$, using the following formula:

$$\hat{N}_u(D) = \frac{M_i^{(j+1)/(j-i)}}{M_j^{(i+1)/(j-i)}} \hat{h}(x). \tag{5}$$

In summary, we aim to derive a single $\hat{h}(x)$ function, valid for the whole Switzerland and possibly beyond, that can be used to approximate measured HSNDs using 2 of their empirical moments. In practical terms, the $\hat{h}(x)$ fitted over the training set of hail sensor measurements is used to compute $\hat{N}_u(D)$ over all the datasets available, including the HSND derived from the
drone observations.

### 3.2 Error metrics

In this section, we introduce a series of metrics to evaluate how closely an estimated value ($\hat{V}$) matches a reference one ($V$). Examples of such comparisons include evaluating the similarity between $\hat{N}_u(D)$ and $N_u(D)$, or estimating the differences between the values of $h(x)$ in each of the three regions of deployment of the hail sensors. Given the similarities between our
analysis and the one of Raupach et al. (2019), we will follow the latter in defining the error metrics used in our analysis.

The quantities $V$ and $\hat{V}$ depend on the object of the comparison. When dealing with distributions, such as $N_u(D)$ and $h(x)$, the variables $D$ and $x$ undergo a binning procedure, which divides them into equal intervals, to which an index $k=1,...,K$ is assigned. Note that in the computation of the error metrics, the index $k$ spans the interval of diameters between the smallest and





largest hailstone recorded for the event taken into consideration. At each bin $D_k$ (or $x_k$), the (normalized) distribution will have

a single value $N_u(D_k)$ $(h(x_k))$. The latter corresponds to $V_k$ in the definition of the error metrics presented in the following formulas, while the reconstructed $\hat{N}_u(D_k)$ $(\hat{h}(x_k))$ is used as $\hat{V}_k$. However, in section 4, the same error metrics are used for the comparison of a different set of quantities: the moments of the distributions. In that specific case, the index $k = 1, ..., K$ indicates the numbering of the hail events in the dataset, while $V_k$ and $\hat{V}_k$ represent the measured and estimated moments of the HSND for that specific event.

Having defined the meaning of $V_k$ and $\hat{V}_k$, we can proceed with the introduction of the first two error metrics:

- the bias, computed as:

$$\text{bias} = \frac{1}{K} \sum_{k=1}^{K} (\hat{Y}_k - Y_k),$$

- the root mean square error (RMSE), defined as:

$$\text{RMSE} = \sqrt{\frac{1}{K} \sum_{k=1}^{K} (\hat{Y}_k - Y_k)^2}.$$

While RMSE gives us information on the overall error between the estimated and reference distributions, the bias provides some additional insight into potential underestimation or overestimation. Both metrics share the same units, which depend on the quantities that are compared, and which will be specified throughout the text.

The third metric is the relative bias, expressed as a percentage, and defined as:

$$\text{Relative bias} = 100 \cdot \frac{\hat{V}_k - V_k}{V_k}$$

Only non-zero values of $V_k$ can be used for the computation. Differently from the previous metrics, the relative bias provides us $K$ values, one for each index $k$.

The last metric used in this study is the Pearson correlation coefficient (PCC), which is unitless. This quantity gives us a measure of the linear correlation between the reference values and the estimated ones.

## 4 Selection of the HSND moments

As anticipated in Sections 2.1.1 and 2.1.2, there are a series of factors that can impact the number of hailstones recorded at the two ends of the HSND. One of them is the issue with the smallest detectable diameter, which may lead to the exclusions of some potentially valid measurements from the dataset. This effect is most noticeable for diameters below 7 mm. Another factor affecting $N_u(D)$ is the choice of a threshold on the minimum number of hailstones in an event, which can result in the exclusion of some of the largest diameters. Furthermore, due to the rarity of big hailstones, the randomness of the impacts on

the relatively small surface of the hail sensor has the potential to be noticeable in the upper tail of the HSND.



In this section, we examine the effect of these missing measurements on the value of the empirical moments computed for each event. As described in section 3.1, the double moment normalization requires the selection of two orders, denoted by $i$ and $j$. The current analysis allows us to choose the values of $i$ and $j$ to limit, as much as possible, the impact of missing sections of the HSND on $M_i$ and $M_j$.

The empirical moments of orders between 0 and 6 have been computed for all the events recorded by the hail sensors, providing us with a set of reference values. The computation of the moments is repeated over 3 sets of modified HSNDs. Two types of changes are implemented:

  – removal of hailstones with relatively small diameters, defined as the ones with $D < 7\,\mathrm{mm}$,

  – removal of hailstones with a relatively large diameter, comprising all diameters above the quantile 0.9 of the HSND for
each event.

The first two sets have been created by applying the two conditions separately and, in the third one, the two have been used at the same time. Using the error metrics introduced in Section 3.2, the moments from the original set of HSNDs are compared to the ones from the modified distributions, and the results are displayed in Figure 3.

    The relationship between the orders and the value of bias and RMSE has a straightforward explanation. The double mo-
ment normalization, in the formulation presented in section 3.1, requires the usage of the non-normalized empirical moments. Therefore, the value of the moments used for the current analysis increases considerably with the order, since the latter is used in the exponentiation of the value of the diameters. A comparison between high-order moments will likely yield differences that are larger, in absolute value, than their low-order counterpart. Nevertheless, panels 3.a and 3.b provide some useful insight into how the removal of small or large hailstones affects the value of the empirical moments. A comparison with the moments
computed after removing both (blue bars in the figure) reveals that the exclusion of diameters below $7\,\mathrm{mm}$ (light gray bars) is likely the largest contributor for the lowest orders. On the contrary, the removal of diameters larger than the quantile 0.9 of the HSNDs (dark gray bars) dominates the error at the highest orders.

    Instead of increasing monotonically like the bias and RMSE, the relative bias (panel 3.c) and the Pearson correlation co-efficient (panel 3.d) reach their optimum at the orders 2 and 3. This behavior can be explained by the combined effect of
removing small and large hailstones, affecting more noticeably the low and high moment orders, and having a smaller impact on the orders in the middle. Similarly to the bias, the relative bias is always negative in this comparison, since the removal of a certain number of impacts from the HSND necessarily results in a lower value for the empirical moments. The latter can be considerably lower than its counterpart computed from the original HSND, as shown in panel 3.c, where the relative difference is below -50% in most cases. The difference becomes slightly more noticeable at the highest orders than at the lowest ones.
The correlation coefficient behaves similarly, with moments 2 and 3 having PCC values above 0.9, while moment 6 does not even reach a correlation of 0.5. In this case, the PCC of the moment of order 4 is considerably lower than the one visible for order 1.

    The information that these four error metrics give us can be used to select the pair of orders $[i, j]$ to be used for the double moment normalization. In the case of its application to drop size distributions, Lee et al. (2004) highlights the importance of





choosing two non-consecutive orders. Despite our analysis being centered on hail, rather than rain, we will follow the same indication. According to the values of relative bias and PCC shown in Figure 3, the orders that are the least affected by potential missing diameters at the two ends of the HSND are the ones between 1 and 4, with the best values visible at order 2 and 3.

We selected the order pair $[2,4]$ for performing the rest analysis. Results for all other pairs can be found in Appendix A. The latter shows issues in the fit of $\hat{h}(x)$ for the pair $[1,3]$, which explains our choice of $[2,4]$ despite the order 4 being potentially

more negatively affected than the order 1.

## 5 Normalized distributions

The main benefit of computing a normalized distribution is the reduced spread of the $h(x)$ values at each $x$ when compared to the considerable variability of $N_u(D)$ at each $D$. This effect, sometimes referred to as the "collapse" of the normalized distributions, is what allows us to fit $\hat{h}(x)$, as described in section 3.1. Throughout this section, all normalized distributions

have been computed at discrete values of the normalized diameter $x$. The range of $x$ values has been divided into intervals $dx = 0.1$ and, in the text, we will refer to each of these intervals by the value of their center point (e.g. $x = 0.85$ indicates the values between $x = 0.8$ and $x = 0.9$).

### 5.1 Definition of training and test set

As mentioned in Section 3.1, the measurements collected by the network of hail sensors have been split into a training and

a test set. While the former provides us with the data to compute $\hat{h}(x)$, the latter is used for the evaluation of the fit of the normalized distribution and, later in the text, of the similarity between $\hat{N}_u(D)$ and $N_u(D)$. The training and test sets contain approximately 70% (67 events) and 30% (27) of the whole hail sensor dataset, respectively. Data are assigned to the two sets using the Python pseudo-random number generator (Python Software Foundation, 2023) and following a simple set of rules: no hail sensor location can appear in both sets, and the data from the three regions (Jura, Napf, Ticino) must follow the same

70%-30% split.

Figure 4 shows the values of $h(x)$ from the training (panel 4.a) and test (panel 4.b) sets. Given the presence of tens of events in each of them, we decided to display the relationship between $x$ and $h(x)$ as a 2-dimensional histogram instead of individual curves. The highest number of counts in the histogram are clustered in a relatively narrow band, which follows a negative trend, with the highest values of $h(x)$ visible at the smallest normalized diameters.

### 5.2 Fit of the normalized distributions

Once the value of $h(x)$ has been computed for all the events in the training set, we can fit $\hat{h}(x)$. In practical terms, the fit is performed by minimizing the RMSE between $\hat{h}(x)$ and the values of $h(x)$ from individual events in the training set. However, the RMSE can be sensitive to a few isolated points, if they deviate considerably from the narrow band of values that we are attempting to capture with the fit. In Figure 4, examples of such points can be found at the lowest and highest $x$. Therefore,





the fit has been limited to the discrete $x$ for which at least five valid $h(x)$ values have been recorded in the training set. This condition effectively limits the interval of suitable $x$ between $x = 0.45$ and $x = 1.75$.

The fit has been repeated with no limits on the minimum number of valid $h(x)$, to assess its impact on the final product. While the differences in $\hat{N}_u(D)$ between the two cases are minor, limiting the $x$ interval has some small benefit in improving the performances for higher diameter sizes. A larger threshold (equal to 10) on the number of valid $h(x)$ has also been tested,

but the excessive limitation that it imposes on the range of suitable $x$ results in the fitting procedure failing to find optimal values for $\mu$ and $c$.

With the chosen condition in place, the values of the two parameters resulting from the fit are $c = 0.37$ and $\mu = 46$, and the curve associated with them has been displayed as a continuous black line over the normalized distributions of the training (panel 4.a) and test (panel 4.b) sets. The gamma model follows the region with the highest counts in the 2-dimensional

histograms, especially for $x$ close to 1.

The error metrics introduced in section 3.2 provide us a way to quantify the similarity between $\hat{h}(x)$ and $h(x)$. Their values are displayed at the two leftmost positions in the four panels that compose Figure 5. The error metrics have been computed separately for each event, resulting in tens of values for each set. We decided to summarize this information by showing a series of statistics of their distribution. The type of visualization used is known as "boxplot", which comprises a horizontal

line indicating the median value, a box representing the interquartile range (IQR), two capped lines to show the position of the quantiles 0.05 and 0.95, and all values outside this interval are displayed as circular markers. Since the relative bias provides us with a series of values for each event, instead of a single number, we decided to show only the distribution of its mean for each event in the training and test sets.

Panel 5.a reveals the predominance of negative values for the bias, which are consistent with the position of the $\hat{h}(x)$ curve

in Figure 4, slightly below the region with the highest count numbers in the histogram. The bias over the training and test set have similar median values, while a more marked difference can be observed in the RMSE, in panel 5.b. Both metrics are likely affected considerably more by the errors at the smallest normalized diameters, for which $h(x)$ is orders of magnitude larger than at the highest $x$ values. While the generalized gamma model struggles to fit adequately the values for $x < 0.5$, we could not find a suitable alternative that results in better overall performances. From a visual inspection of Figure 4, it may seem

that an exponential model for $\hat{h}(x)$ may better capture the trend in the data points. However, given the conditions defining the $\hat{h}(x)$ candidates suitable for the double moment normalization, in the current formulation the only possible exponential model would be the one obtained by replacing $\mu = 1$ and $c = 1$ in equation 4. We tested this possibility, and while it reduces the underestimations for the smallest normalized diameters, it also results in a slightly higher underestimation for the remaining $x$ interval, while also lowering the overall performances in all other error metrics.

Returning to the analysis of Figure 5, the relative bias is centered around 0%, but the location of the quantiles is asymmetrical, with the quantile 0.05 closer to the median than the quantile 0.95. More variability can be observed in the training set than in the test set, probably due to the higher number of events present in the former. Finally, the Pearson correlation coefficient behaves similarly to the RMSE, with the training set having a better score than the test set. While the former has a median value close to 0.9, the latter has a median correlation of approximately 0.8 and a larger interquartile range.




Given the nature of $x$ and $h(x)$, the exact value of some of the metrics can sometimes be difficult to interpret. However, they allow us to understand better the reason behind some of the differences between $\hat{N}_u(D)$ and $N_u(D)$, discussed later in the text. Additionally, they provide a reference for the comparison with the drone-derived dataset, presented in the next section.

### 5.3    Comparison with the drone-derived normalized distribution

The normalized distribution computed from the HSND retrieved by the drone is shown in Figure 4 as a series of orange circles
displayed above the 2-dimensional histogram and alongside the $\hat{h}(x)$ line. Despite the difference in $N_u(D)$ between drone and hail sensors, shown in Figure 2, the alignment between the $h(x)$ from the drone and the region with the highest counts in the 2-dimensional histogram is remarkable, especially around $x = 1$. The normalized distribution of the drone event follows the $\hat{h}(x)$ line, too, even though none of its measurements has been used for the fit. Some discrepancies can be seen around $x = 0.5$ and $x = 1.5$, where the $h(x)$ values from the drone dataset are within the extremes of the populated bins of the 2-dimensional
histogram, but markedly above the $\hat{h}(x)$ line. The drone $h(x)$ is again closer to the normalized distribution fit at the highest $x$ values, where only a handful of measurements from the hail sensors are available.

        In terms of error metrics, a comparison of $\hat{h}(x)$ with the drone-derived normalized distribution is provided in Figure 5, alongside the metrics for the training and test sets. A negative value for the bias can be seen in the drone dataset, similar to the one described in section 5.2, and consistent with the location of the fit line with respect to the $h(x)$ markers in Figure 4. In most
cases, both bias and RMSE for the drone dataset are higher (in absolute value) than their hail sensor counterparts, and close to the quantiles 0.05 (for the bias) and 0.95 (for the RMSE) of the distributions for the training and test sets. Contrasting with some extremely positive values in the relative bias in the hail sensor dataset, the comparison of $\hat{h}(x)$ with the drone-derived $h(x)$ results in a negative relative bias, once again close to the quantile 0.05 of the distributions from the training and test sets. The Pearson correlation coefficient is the only metric in which the fitted $\hat{h}(x)$ has better performances for the drone event than
for most of the hail sensor ones.

        Despite $\hat{h}(x)$ having better skills overall in approximating most of the hail sensor events than the drone one, its similarity to the drone-derived $h(x)$ is still notable. These results offer us a first confirmation of the validity of applying the double moment normalization to model the HSND, especially given the independent nature of the drone measurements and their coverage of diameter ranges rarely recorded by the hail sensors. However, the ability of a unique $\hat{h}(x)$ to correctly represent the normalized
distribution throughout the Swiss territory still needs to be tested. This verification of the spatial invariability of the normalized distributions is presented in the next section.

### 5.4    Spatial invariability of the normalized distributions

While the previous analysis was based on a comparison of the fitted $\hat{h}(x)$ with the values of $h(x)$ derived from drone and hail sensor measurements, in this section we present a direct comparison of normalized distributions from the three regions of
deployment of the hail sensors. By looking at how much the $h(x)$ values from each region resemble the ones from the same and other regions we can investigate whether significant differences between them exist.





The first comparison has been performed on an event-by-event basis, and the value of the resulting error metrics are shown in Figure 6. For each event in a reference region, indicated on the x-axis of the figure, its normalized distribution is compared to $h(x)$ from a different event, whose geographical region is indicated by the color of the boxplot. Given the high number of

possible combinations, the spread of values in each metric is larger than in previous figures of the same type.

In all regions, the bias is within the interval $[-1, +1]$ for the majority of the events. Ticino stands out for its higher bias values when compared to the other two regions. The RMSE distributions have median values close to 1, with almost the totality of event pairs having values between 0.1 and 10. Lower RMSE can be consistently seen when Jura or Napf are the reference regions, while Ticino often has the highest ones, with some remarkable extreme values. An opposite pattern can be observed

for the relative bias, which is lower (and often negative) when Ticino is the reference. While a large majority of event pairs have a relative bias between -50% and 50%, there are some exceptional values, reaching up to 1000%, especially when Ticino is the comparison region. The Pearson correlation coefficient varies considerably, with the Jura once again offering the best score when used as the reference. Negative correlations are visible in all cases, even though they represent a minority of all event pairs.

Despite the regional differences listed above, the similarity in the error metrics among the different combinations is noticeable. In all cases, the median value of the metric for event pairs belonging to the same region is within the interquartile range of the values for events from different regions. Additionally, the boxplots show considerable overlap between the three IQRs (and between the interquantiles $[0.05, 0.95]$ ) at each reference region, in all four panels. These results suggest that the variability of the normalized distributions of different events within the same region is comparable to the variability between events from

different regions. Together with the similarity between the $h(x)$ computed from the drone dataset and the hail sensors one, this result is another indicator of the potential suitability of the double moment normalization for modeling HSNDs over the Swiss territory.

Before proceeding with the discussion on the estimated distribution of hail sizes, we include a short comparison between the median value of $h(x)$ at each $x$ value (discretized at a resolution $dx = 0.1$) from the three regions. A similar quantity has been

used by Raupach et al. (2019) to fit $\hat{h}(x)$ when modeling the distribution of drizzle diameters. While in our analysis the fit has been performed over all events from the training set at once, without pre-computing any statistics from the original $h(x)$, we decided to include a short comparison of these median values for completeness, and for the clarity with which they highlight even small differences between the regions. The resulting error metrics have been shown in Figure 7, which follows a structure analogous to Figure 6.

Two features of this figure are particularly relevant to our analysis. The first of them is the overall "goodness" of all metrics in Figure 7 when compared with the ones in Figure 4. Our interpretation of these results is that the differences between the typical normalized distribution in the three regions (represented by the median $h(x)$) are smaller than the ones between the individual $h(x)$ values and the generalized gamma model fitted through them.

The second interesting feature is the similarity between the hail events recorded in Napf and the Jura. Even though the

differences between regions are relatively small, Ticino stands out for its worse error metrics in Figure 7. In our opinion, while discrepancies between the typical normalized distributions in the three regions are too faint to be captured by the model we





selected as $\hat{h}(x)$, it may be possible, with a larger dataset, to refine the analysis and explore these spatial differences in more detail.

## 6 Hail size number distributions

In this section, we examine the differences between the distributions $\hat{N}_u(D)$ reconstructed using the double moment normalization and the ones retrieved by the hail sensors and by the drone. The analysis mirrors the one of section 5, with a focus first on the hail sensor measurements and later on the drone ones. In both cases, $\hat{N}_u(D)$ is always computed using $\hat{h}(x)$ presented in section 5.2 and the empirical moments computed from the HSNDs that compose the various datasets, following the formula in equation 5. The value of $\hat{h}(x)$ has been computed on a fixed series of discrete $x$ values, separated by an interval $dx = 0.1$.

Since the conversion between $x$ and $D$ requires the value of the empirical moments $M_i$ and $M_j$, the discretization of $D$ varies between events. The resulting size of each bin is usually between $0.5 \, \mathrm{mm}$ and $1 \, \mathrm{mm}$, but smaller and larger values can be observed in some cases.

To better visualize the shape of the HSNDs discussed in the next sections, we include here four examples of the $\hat{N}_u(D)$ from different events, superimposed to the measured $N_u(D)$. These examples, shown in Figure 8, have been recorded by the drone

(panel 8.a) and by the hail sensors in the three regions: Jura (panel 8.b), Napf (panel 8.c), and Ticino (panel 8.d). The events displayed in the last three panels all belong to the test set and have been selected to exemplify the variety in the number of hailstones. In particular, panel 8.d shows an event with a relatively high number of impacts, a condition that is not uncommon in Ticino, while panel 8.b illustrates an example with a much lower number. The event in panel 8.c represents a middle ground between the previous two.

The similarity between $\hat{N}_u(D)$ and $N_u(D)$ for the drone event is noteworthy, especially for diameter ranges between $10 \, \mathrm{mm}$ $< D < 20 \, \mathrm{mm}$ and for $D$ close to $30 \, \mathrm{mm}$. Since a different instrument (hail sensors) has been used for defining $\hat{h}(x)$, this result suggests that $\hat{h}(x)$ is largely linked to meteorology, not the instrument. It should be noted, however, that the reconstructed HSND underestimates the number of impacts, and the same behavior can be noticed for the hail sensor examples, in the other three panels. The lack of hailstones is mainly noticeable at low diameter values, even though the relationship between $\hat{N}_u(D)$

and $N_u(D)$ for $D$ close to $5 \, \mathrm{mm}$ can differ, as exemplified by panel 8.c.

### 6.1 Hail sensors dataset

In this section, we focus the analysis on the sole hail sensor measurements. While the three events in Figure 8 give us some indications on how $\hat{N}_u(D)$ matches $N_u(D)$, a comparison of the two sets of HSNDs encompassing the whole training and test sets require a different type of visualization. Figure 9 provides us two scatterplots, one for each set of hail sensor measurements,

with $N_u(D)$ on the x-axis and $\hat{N}_u(D)$ on the y-axis, and the value of $D$ indicated by the color of each marker. A logarithmic scale has been chosen due to $\hat{N}_u(D)$ spanning several orders of magnitude.

In both the training and test set, a large fraction of the pairs in the scatterplot lie close to the identity line, with more noticeable deviation for low numbers of impacts. While these discrepancies are made more evident by the logarithmic scale, their existence





is linked with $N_u(D)$ being an integer number, while $\hat{N}_u(D)$ can be equal to any positive real one. The underestimation of
the fallen hailstones by $\hat{N}_u(D)$, mentioned in the previous section, appears more systematic for $N_u(D) > 10$, which usually
corresponds to small diameters.

While the figure 9 gives us an overview of the similarity between $\hat{N}_u(D)$ and $N_u(D)$, a quantitative comparison requires
the usage of error metrics, whose value is shown in Figure 10. The value of the bias in panel 10.a confirms again the slight
underestimation of the number of impacts by $\hat{N}_u(D)$, which is in most cases of less than 5 impacts per diameter bin. The
RMSE, in panel 10.b, indicates a slightly higher overall error, whose distribution can reach 10 impacts per bin more frequently
than the bias. The average relative error is centered around 0%, similarly to the $\hat{h}(x)$ case, with an IQR reaching values of
several tens of percent, slightly skewed toward positive values. Finally, the correlation coefficient is relatively high, with a
median value close to 0.9 for the training set and 0.8 for the test set, even though, for the latter, the quantile 0.25 reaches values
slightly below 0.7.

Given the importance of large diameters in the wider context of hail studies, we decided to repeat the analysis including
only impacts with $D > 10$ mm. The error metrics, computed over events with at least 5 hailstones that satisfy this condition,
are shown in Figure 11. Since the largest diameters are relatively rare, the number of events available is low. For this reason,
Figure 11 shows the error metrics for the individual events in the test set, rather than summarizing their distribution with a
boxplot.

The bias values, in panel 11.a, are less negative than the one in panel 10.a, while the RMSE is lower than the one computed
from the full HSNDs. This behavior is not surprising, given the rarity of large hailstones in our dataset. The relative bias
is markedly more negative than its counterpart of panel 10.d, highlighting the underestimation of the number of impacts in
$\hat{N}_u(D)$. The correlation coefficient, instead, is slightly higher, especially for the test set, even though this metric may not be
particularly robust in this case, given the low number of data points available.

In summary, the comparisons presented in this section reveal an overall agreement between $\hat{N}_u(D)$ and $N_u(D)$, with the
former usually underestimating the hail sensor measurements. The differences are particularly noticeable for diameters above
10 mm, where the double moment normalization suffers from a relatively low number of impacts available. In our opinion,
improvements in the representation of this part of the HSNDs may be achieved in future studies. As the hail sensors net-
work continues to operate, more measurements will be available for computing $\hat{h}(x)$ and for a more robust evaluation of the
performances, especially over relatively large diameters.

## 6.2 Drone dataset

The drone dataset is composed of a single event, albeit with a relatively large number of hailstones. Given the differences in
$N_u(D)$ between the drone and hail sensor measurements, the inclusion of the error metrics for the former in Figure 10 would
have altered the scale of the y-axis, making the values associated with the hail sensor datasets difficult to read. Therefore, we
decided to present the error metrics for the drone event separately in Table 1. The table includes values computed over the
whole HSND, as well as the ones for the hailstones whose diameter is larger than 10 mm.





Similarly to the hail sensor case, the double moment normalization results in an underestimation of the drone-derived $N_u(D)$, as indicated by the negative bias value. While both bias and RMSE are considerably larger than their counterparts in Figures 10 and 11, we should keep into account that the drone recorded a particularly large number of hailstones, with
several thousand counts for the smallest diameter bins. This results in a relatively small (and negative) relative bias, whose values decrease slightly for $D > 10$ mm. A close look at panel 10.c reveals a slightly worse performance over the whole drone-derived HSND, when compared to the median relative bias for the hail sensor datasets. However, the performances for diameters above 10 mm are in line with the ones of panel 11.d. Finally, when compared to its hail sensor counterparts, the correlation between $\hat{N}_u(D)$ and $N_u(D)$ for the drone is higher, while also being computed over a larger number of discrete $D$
bins. A small improvement in the correlation coefficient can be seen for $D > 10$ mm.

Overall, while the visual comparison of $\hat{N}_u(D)$ and $N_u(D)$ from panel 2.a shows a good agreement between the two, the error metrics presented in this section highlight some inaccuracies in the output of the double moment normalization. Of particular interest is the underestimation of $N_u(D)$ even for relatively large diameters, which confirms the pattern described in the previous section, with added robustness thanks to the higher number of measurements recorded by the drone. The
relatively high correlation coefficient suggests that the $\hat{N}_u(D)$ follows the shape of $N_u(D)$. So, despite the slightly lower values of impacts in $\hat{N}_u(D)$, we consider these results a positive indicator of the ability of the double moment normalization to scale adequately to a much larger number of impacts than the ones used in the training of the algorithm.

## 7   Summary and conclusions

In this study, the double moment normalization has been used to model the shape of a series of hail size number distributions
collected by the Swiss network of automatic hail sensors. All hail events with more than 30 impacts recorded between 9 September 2018 and 1 September 2023 have been selected, giving us a dataset of 95 HSNDs. An additional HSND, retrieved through the use of drone photogrammetry, has been used to evaluate the method on an independent data source.

According to the theory first presented by Lee et al. (2004), the hailstone diameters ($D$) and their associated number of impacts ($N_u(D)$) are converted into a series of normalized diameters ($x$) and normalized distributions ($h(x)$). After dividing
the hail sensors dataset into a training and test set, we used the former to fit a generalized gamma model ($\hat{h}(x)$). The method, then, uses $\hat{h}(x)$ and two empirical moments of the original distributions ($M_i$ and $M_j$) to reconstruct an estimate of the HSND ($\hat{N}_u(D)$).

Due to variations in the smallest hailstone size detectable by the instrument, the lowest diameter sizes could be erroneously filtered out by some of the instruments for a limited amount of time. Furthermore, the small area in which impacts are detected
and the relative rarity of large hailstones can lead to a certain degree of randomness on how accurately these large diameters are represented in the measured HSND. Therefore, an analysis has been conducted to identify the moments least affected by these potential missing measurements, leading to the choice of the values $i = 2$ and $j = 4$ for the moment orders for the double moment normalization.





Given the deployment of the hail sensors in three separate regions of Switzerland (Jura, Napf, and Ticino), we tested the similarity of the normalized distributions recorded in each of them to verify whether a single $\hat{h}(x)$ fit can accurately represent all of them. A comparison of $h(x)$ from individual events reveals that the variability between events from different regions is comparable to the variability within the region itself. Furthermore, the median values of $h(x)$ from the three regions show a considerable degree of similarity, exceeding the one between $\hat{h}(x)$ and most of the normalized distributions in the training and test set. This behavior indicates a level of uniformity in $h(x)$ over the three regions adequate for our choice of a sole generalized gamma model as $\hat{h}(x)$. However, the relatively high variability in the normalized distribution of the events in Ticino, highlighted by the error metrics in section 5.4, may indicate some differences between this region and the remaining two. As mentioned in the data section, Ticino is the only region south of the Alps, and the one in which the largest hailstones have been recorded. These features could be investigated in future studies, when more measurements will have been collected by the hail sensors.

The fit of the normalized distribution has been evaluated by comparing $\hat{h}(x)$ with $h(x)$ for all events. While $\hat{h}(x)$ is close to the most common values of $h(x)$ at each discrete $x$, it usually underestimates slightly the value of the normalized distribution in both the training and test sets. Despite the considerably higher number of impacts recorded by the drone, its $h(x)$ are close to their hail sensor counterparts and $\hat{h}(x)$. This underlines the reduction in variability resulting from the usage of the normalized distribution rather than the original HSNDs and further confirms the suitability of the double moment normalization to model the hail size number distribution.

The final evaluation of the proposed method lies in a comparison between the reconstructed and original HSNDs, which reveals that $\hat{N}_u(D)$ is overall similar to $N_u(D)$. In the case of the drone measurements, this similarity is particularly striking, considering the difference in the number of impacts between this event and the ones used in the training of the algorithm. However, a negative bias is noticeable in all the datasets analyzed. The RMSE between the reconstructed and original distribution is often between 5 and 10 hailstones per discrete $D$ bin for the hail sensor, while it reaches up to several hundreds in the drone case. This value usually decreases when only relatively large hail stones ($D > 10\,\mathrm{mm}$) are taken into consideration.

In summary, the most noticeable differences between $\hat{N}_u(D)$ and $N_u(D)$ are at the two ends of the HSND. These discrepancies, often consisting of an underestimation of the number of impacts by $\hat{N}_u(D)$, suggest that better performances may be achievable by using a different function for the $\hat{h}(x)$ fit. However, the limited amount of events available in our dataset complicates the testing of complex $\hat{h}(x)$ models. The randomness and variability intrinsic to the hail sensor measurements would undermine the robustness of a hypothetical analysis that strays from previous studies using a completely different model for $\hat{h}(x)$. Fortunately, given the permanent nature of the hail sensor installation, the dataset of hail events grows with every passing year. As this number increases, future studies may be able to experiment with more unusual functions for $\hat{h}(x)$. Additional measurement campaigns with the drone may also provide sets of events with a high number of impacts for a correct evaluation of the reconstructed HSND for relatively large hailstone diameters.

Finally, the results presented in this study offer, in our opinion, an opportunity for future investigation into the retrieval of HSNDs from weather radar. By finding a link between the empirical moments of the HSND and the radar measurements, it would be possible to use the formula and parameters of $\hat{h}(x)$ defined in this study to estimate the full distribution of hail





diameters expected at the ground. This application would be similar to the ones retrieving the drop size distribution in rain

(Raupach and Berne, 2017) or drizzle (Raupach et al., 2019) cases. While the moment's orders 2 and 4 have been chosen for our analysis, other pairs may have clearer links to certain sets of features in the radar data. In this case, Appendix A provides a summary of the performances that could be expected for different combinations of empirical moments.

**Appendix A: Double moment normalization for all moment pairs**

While the main body of the article focuses on the moment orders $[2,4]$, in this appendix we summarize the results for additional

moment pairs. All combinations of orders between 0 and 6 have been tested, resulting in a total of 21 pairs. In the following sections, we present a series of the error metrics to compare $\hat{N}_u(D)$ and $\hat{h}(x)$ with $N_u(D)$ and $h(x)$ respectively, computed over the test set and over the drone dataset.

**A1 Normalized distributions for all moment pairs**

For each pair of moment orders, a generalized gamma is fitted over the normalized distributions of all events in the training

set. The parameters $c$ and $\mu$ resulting from the fit are displayed in Table A1. Due to computational limitations, the range of values for the two parameters had to be limited to $[10^{-6}, 500]$ during the fit, as explained in section 3.1. These limits are visible in the table as the value for $c$ or $\mu$ for the moment pairs $[0,1]$, $[0,2]$, $[0,3]$, $[0,4]$, $[0,5]$, $[1,2]$, $[1,3]$, $[1,4]$, $[2,3]$, $[4,5]$, $[4,6]$, and $[5,6]$. Their appearance may indicate that the chosen model for $\hat{h}(x)$ may not adequately fit the shape of the normalized distribution for these specific moment pairs. This analysis complements the one presented in section 4, highlighting how some

combinations of moments less sensitive to the lack of measurements at low and high $D$ (e.g. $[1,3]$) may still not be suitable for the proposed method (and $\hat{h}(x)$ model).

The values of $c$ and $\mu$ are used to compute $\hat{h}(x)$ for a series of discrete normalized diameters $x$ (resolution $dx = 0.1$), which is compared to the normalized distribution $h(x)$ of each event in the test set. The distributions of values for the four error metrics introduced in section 3.2, computed for each moment pair, are displayed in Figure A1. The moments $[2,4]$, used

throughout the study and highlighted in blue in the figure, have average performances when compared to the other pairs. Their values of relative bias are noticeable for their closeness to 0%, while their $PCC$ is lower than the one visible for many other moment combinations. Among the other pairs, the ones including the orders 0 and 1 have bias and RMSE values particularly close to 0, even though their associated $\mu$ parameter may indicate some issue during the fit, as explained above. High orders, on the other hand, have the worst performances in terms of bias and RMSE, with often a markedly broad distribution for the

two metrics.

A similar comparison between $\hat{h}(x)$ and $h(x)$ for the drone dataset has been performed, and the resulting error metrics are displayed in Figure A2. Panel A2.a and A2.b illustrate a behavior similar to the one described for Figure A1, with combinations including high moment orders usually having worse performances than their low-order counterparts. A pattern emerges from panels A2.a, A2.b, and A2.c: the skills of $\hat{h}(x)$ are better the closer the moment orders in the pair are. A noticeable exception

can be found for the pair $[0,2]$ in panel A2.c, which stands out for its extremely high relative bias. Finally, the pattern in


panel A2.d is different from the other panels. The correlation coefficient is slightly lower for pairs including the lowest and highest moment orders, and the maximum value is instead achieved by the pair $[2, 4]$.

## A2  Hail size number distributions for all moment pairs

Having computed $\hat{h}(x)$, it is possible to estimate the HSND for each event in the test set using equation 5. The resulting $\hat{N}_u(D)$ is compared to $N_u(D)$, giving us the distribution of error metrics presented in Figure A3. While the bias shares some similarity to its $\hat{h}(x)$ counterpart in panel A1.a, the differences between different moment pairs are less noticeable. This behavior is even more evident in panel A3.b, in which high moment orders do not have markedly higher RMSE values than the low-order ones. In both cases, the pair $[2, 4]$ has performances in line with most other moment combinations. Panel A3.c also shares some similarities with panel A1.c, with the moments $[2, 4]$ having a relative bias centered around 0%, even though there are a few other pairs with similar skills, and an even narrower distribution of the error metric. In the case of the correlation coefficient, shown in panel A3.d, the median value is often between (or close to) 0.8 and 0.9 for all moment combinations. The pair $[2, 4]$, as in the $\hat{h}(x)$ case, has one of the worst performances, with a relatively low median and a large IQR.

The same error metrics, computed for the drone dataset, are shown in Figure A4. Interestingly, the bias and RMSE deviate from the $0 \, \#/\mathrm{bin}$ value more rapidly as the values of $i$ and $j$ increase than in the case of normalized distributions described in the previous section. The pattern described for panels A2.a, A2.b, and A2.c can be seen also in Figure A4, with combinations of moments of similar order outperforming the ones in which the difference between $i$ and $j$ is larger. The values of the correlation coefficient in panel A4 are almost identical to the ones in panel A2.d, and the pair $[2, 4]$ is once again the one with the maximum $PCC$ value.

Overall, the comparison between the various combinations of moments does not show any of the pairs clearly outperforming the others. In terms of HSNDs and normalized distributions, the choice of $i > 3$ (with $j > i$) often results in worse values of bias and RMSE. While lower orders are often associated with better performances, Table A1 highlights how difficulties can be encountered in their $\hat{h}(x)$ fit. The comparison between different moment combinations, in conjunction with section 4, suggests that the pair $[2, 4]$ is a valid choice for the analysis presented throughout the article: for this pair, the $\hat{h}(x)$ fit does not results in extreme $\mu$ and $c$ values, while the error metrics associated with $\hat{N}_u(D)$ are in line with the one seen for several other combinations of moments, with a few cases in which they excel (e.g. $PCC$ for the drone dataset).

*Author contributions.* AF, JK, ML, MG, UG, and AB designed the study. JK provided the code and information needed for the processing of the hail sensor dataset. ML collected and processed the drone measurements. AF performed the analysis. AF prepared the manuscript, with contributions from ML, JK, MG, UG, and AB., and supervision from MG, UG, and AB.

*Competing interests.* AF, JK, ML, MG, and UG declare that no competing interests are present. AB is associate editor for AMT.



635   *Data availability.* Until the end of the operation of the hail sensor network, La Mobilière is the data owner of the hail sensor measurements. At the end of the operation, the data ownership goes from La Mobilière to MeteoSwiss. For the time being, we have to refer any request for data to La Mobilière.

   *Acknowledgements.* We would like to acknowledge La Mobilière for the funding and inNET for the technical implementation of the network of 80 hail sensors. We would also like to thank all the MeteoSwiss and EPFL-LTE collaborators for their support, and in particular Anne-
640   Claire Billault–Roux and Gionata Ghiggi for their input on the double moment normalization, and Alessandro Hering for his help in accessing the hail sensor data.



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



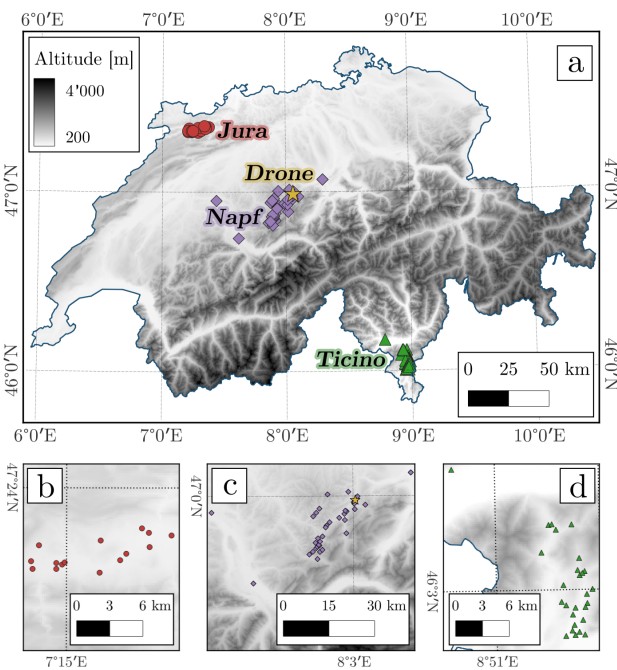

**Figure 1.** The location of the three groups of hail sensors displayed on the elevation map of Switzerland. The groups are shown as different markers: red circle for the Jura, purple diamonds for Napf, and green triangle for Ticino. A yellow star indicates the location in which the drone measurements have been collected. While panel a provides an overview of the entire Swiss territory, the details of each region are provided in the bottom row of panels: panel b for the Jura, panel c for Napf, and panel d for Ticino. The digital elevation model used in all panels has been provided by the Federal Office of Topography Swisstopo (2021).



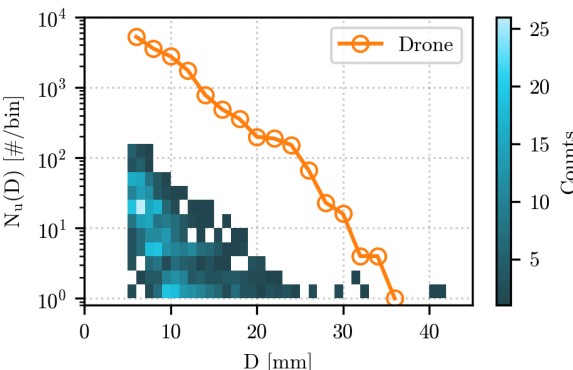

**Figure 2.** Hail size number distributions from the hail sensors and the drone. The number of impacts for each diameter value (resolution of 1 mm) in all events recorded by the hail sensor is shown as a 2-dimensional color bar, with the conversion between color and counts provided by the color bar on the right of the panel. The number of impacts recorded by the drone is shown as an orange continuous line for diameters above 5 mm (included in the analysis) and as a gray dashed line for the ones below 5 mm (excluded from the analysis).



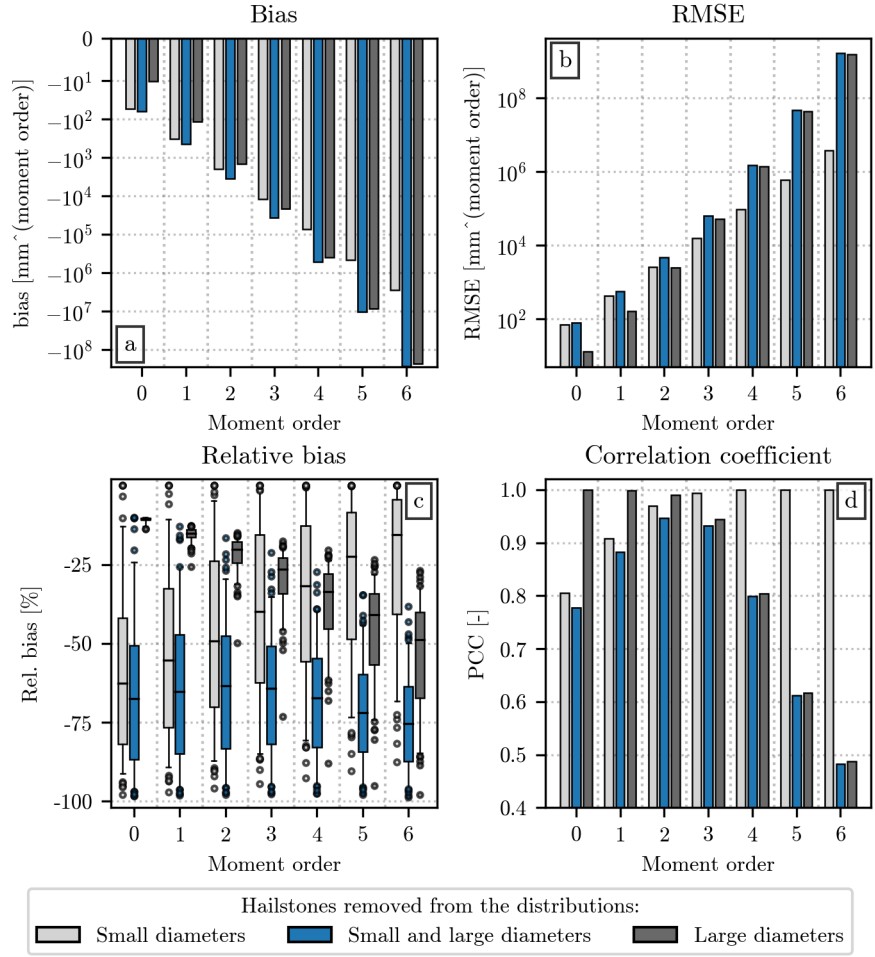

**Figure 3.** Comparison of the value of the empirical moments (orders 0 to 6) computed from the original HSND and from the ones in which small and/or large hailstones have been removed. The analysis uses measurements from all the events of the dataset. Panel a shows the bias between the two sets of moments, panel b the RMSE, panel c the relative bias, and panel d the Pearson correlation coefficient. The moment order is displayed on the x-axis, while the value of the error metric is on the y-axis. In panels a, b, and d, the height of each bar indicates the value of the metric, while in panel c the relative bias is shown as a boxplot, in which its median value is indicated by the central horizontal line, the box shows the interquartile range (IQR), the capped vertical line represents the location of the quantiles 0.05 and 0.95 and values outside of this range are shown as circular markers. In each panel, the color of the bar/box indicates the HSND used for the comparison: light gray ones indicate the HSND in which only impacts with $D < 7$ mm have been removed, the dark gray ones refer to the HSND in which the diameters from the largest 10% of the distribution have been removed, and the HSND resulting from the combination of the previous two conditions are shown by the blue bars/boxes.

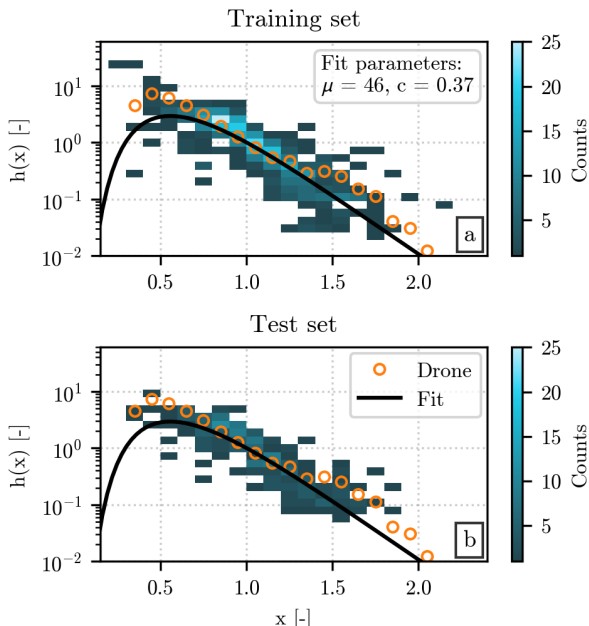

**Figure 4.** Normalized distributions from the hail sensors and drone measurements. The value of $h(x)$ at each discrete normalized diameter (intervals $dx = 0.1$) for the events in the training (panel a) and test (panel b) sets recorded by the hail sensors are shown as a 2-dimensional histogram, with the conversion between color and counts provided by the color bars on the right of the panels. The fitted generalized gamma is represented by a black continuous line, and the parameters $\mu$ and $c$ are shown at the top right corner of panel a. The value of $h(x)$ for the drone event is shown as orange circles.



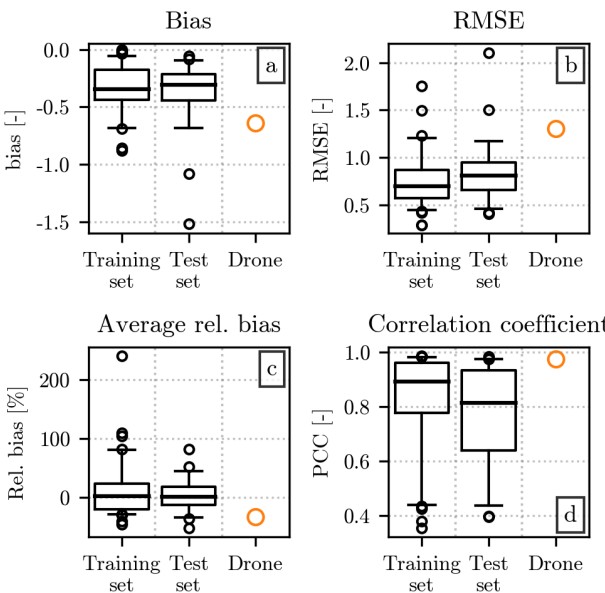

**Figure 5.** Comparison of the fitted $\hat{h}(x)$ with the normalized distributions in the training and test set, and with $h(x)$ computed from the drone measurements. Each panel shows a different error metric, following the same order as Figure 3. The values of each metric in the training and test set are shown as boxplots, with a structure analogous to the one used for the relative bias in Figure 3. The values of the error metrics for the drone event are shown as an orange circle.

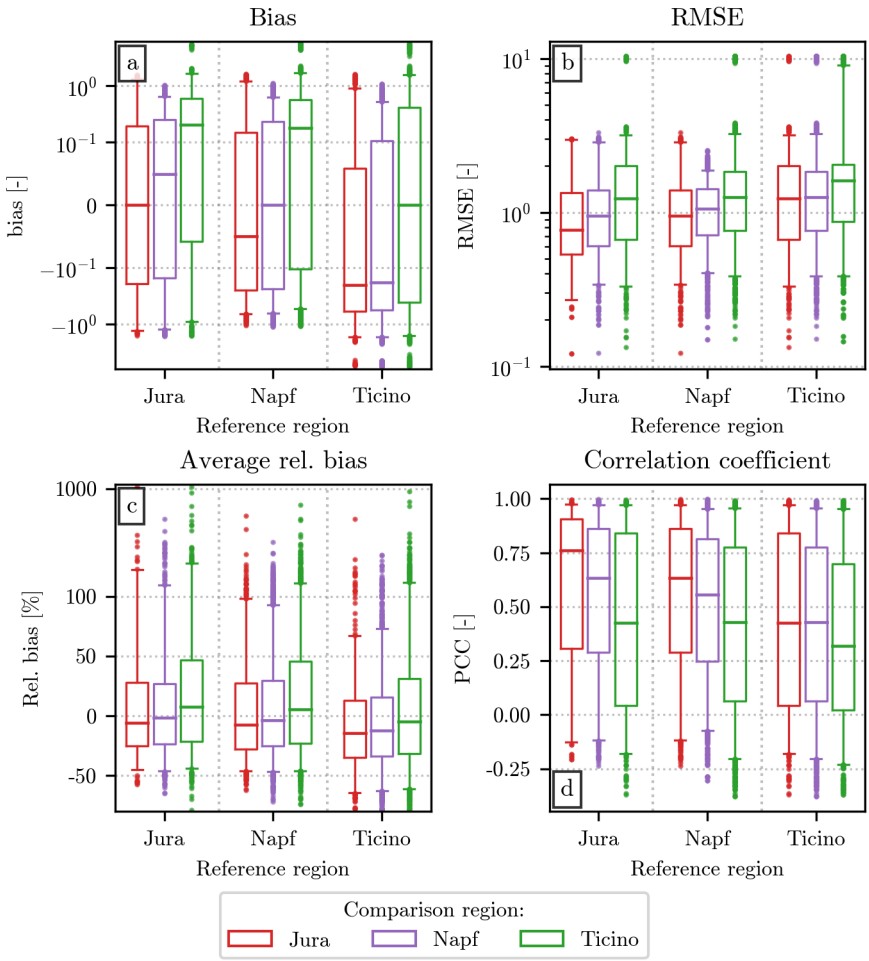

**Figure 6.** Comparison of the normalized distributions from all the events recorded in each of the three regions of deployment of the hail sensors. The comparison is always performed between pairs of events, one of which belongs to a reference region shown on the x-axis, while the region of the second one is shown by the line colors and the position of the boxplot: red for the Jura (leftmost one), purple for Napf (central one), and green for Ticino (on the right). All combinations of different events have been used for the computation of the error metrics. The order in which these error metrics have been assigned to different panels and the structure of the individual boxplots is analogous to the one in Figure 5. The scale of the y-axis in panel a and d is linear inside the intervals [-0.1, 0.1] and [-100, 100], respectively, and logarithmic outside of them.



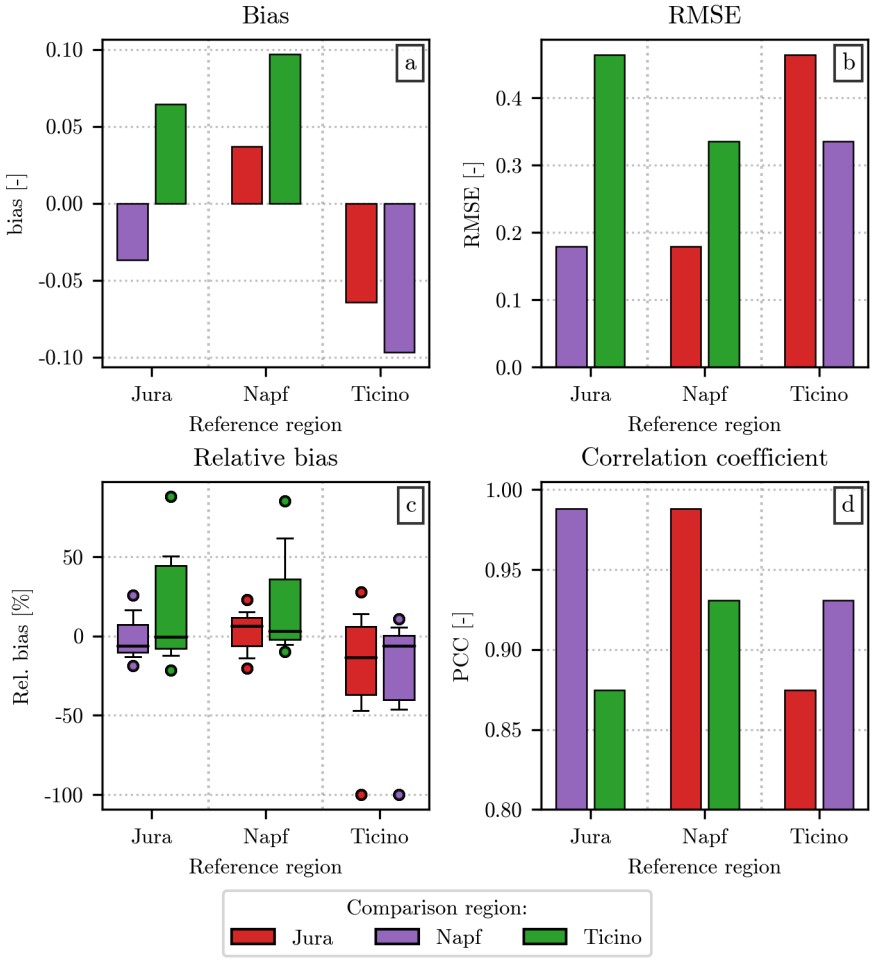

**Figure 7.** Comparison of the median value of $h(x)$ between the three regions of deployment of the hail sensors. Similarly to Figure 3, the values of bias (panel a), RMSE (panel b), and PCC (panel d) are indicated by the height of the vertical bars, while the information on the distribution of the relative bias (panel c) is provided in the form of boxplots. As in Figure 6, the reference region for $h(x)$ is shown on the x-axis, while the color of the bars/boxplots shows the region to which it is compared.



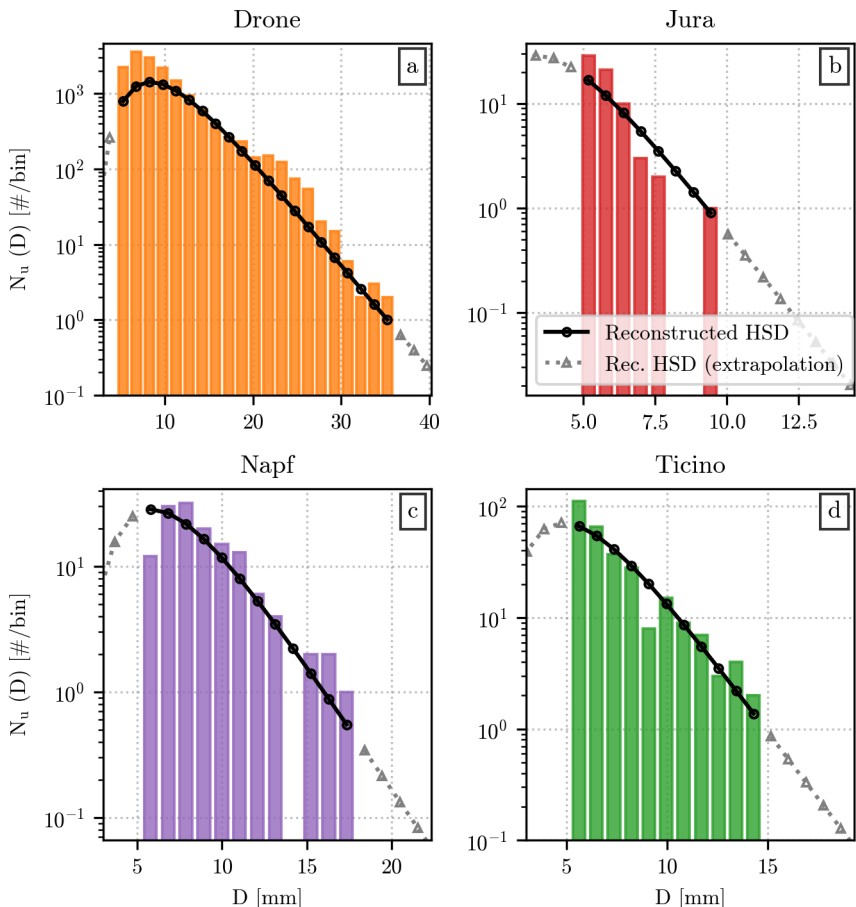

**Figure 8.** Original and reconstructed HSNDs from the hail sensors and drone measurements. Each panel shows the value of $N_u(D)$ as vertical bars, with the color indicating their origin: drone measurements in orange (panel a), Jura in red (panel b), Napf in purple (panel c), and Ticino in green (panel d). For the three regions, only a single event from the test set has been displayed. The HSND reconstructed using $\hat{h}(x)$ and the moments from the original HSND are shown as black lines, with circular markers indicating the exact value at the position of the bars. The value of the reconstructed HSND beyond the minimum and maximum measured diameters is shown as a gray dotted line, with triangular markers.

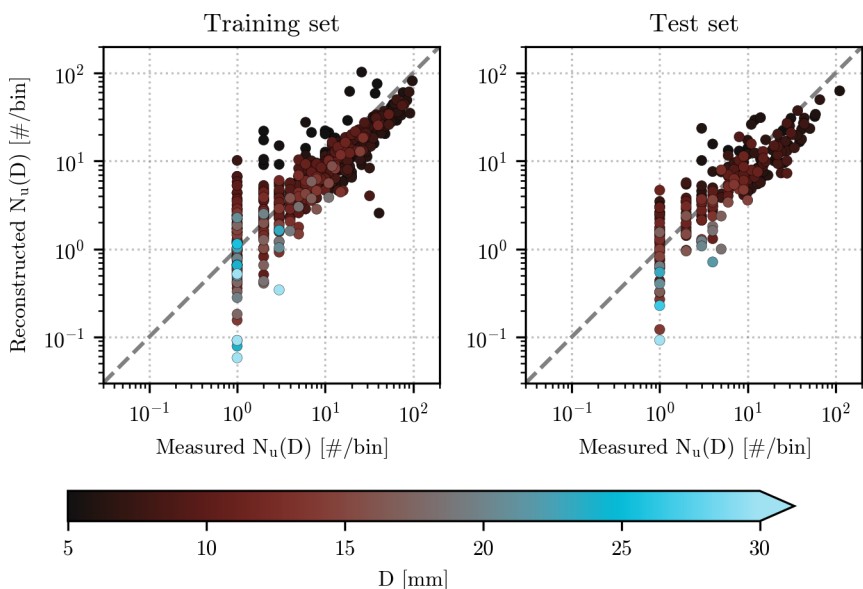

**Figure 9.** Comparison of $N_u(D)$ (on the x-axis) and $\hat{N}_u(D)$ (on the y-axis) over the training (panel a) and test (panel b) sets. The value of the diameter $D$ corresponding to each pair $[N_u(D), \hat{N}_u(D)]$ is shown by the color of the circular markers, while the corresponding color bar has been placed at the bottom of the figure. The gray dashed line that divides each panel represents the identity line.



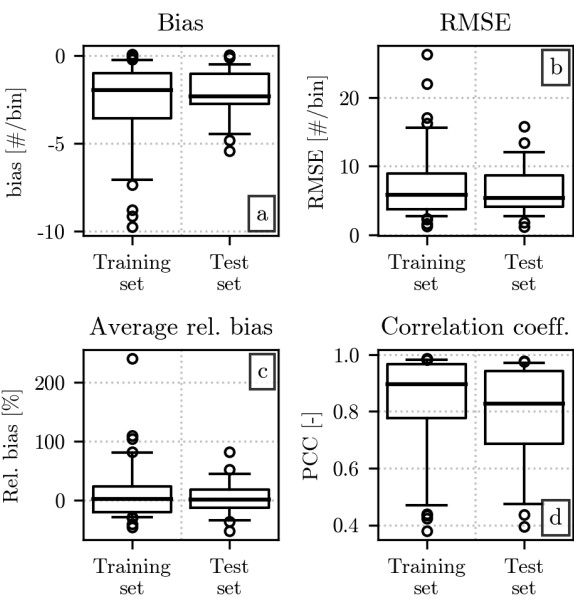

**Figure 10.** Error metrics from the comparison of $N_u(D)$ and $\hat{N}_u(D)$ over the training and test set. The placement of the error metrics in each panel and the structure of the individual boxplots is the same as in Figure 6. While the training and test set are shown at different positions on the x-axis, the drone HSND has been excluded from the figure due to its considerably higher value of bias and RMSE.



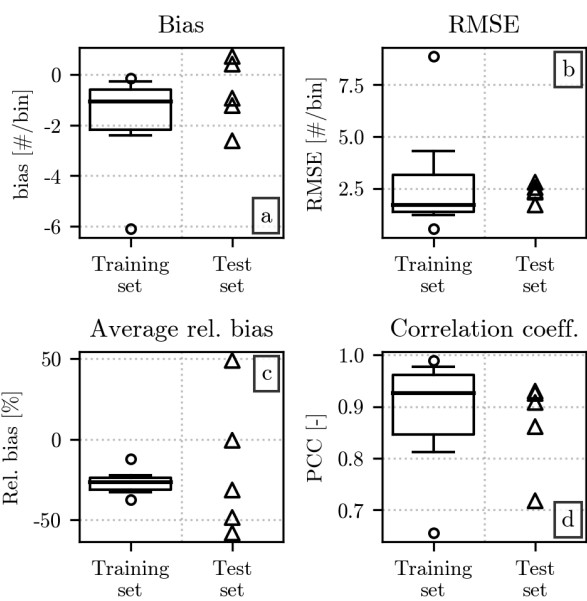

**Figure 11.** As Figure 10, but using only the values $N_u(D)$ and $\hat{N}_u(D)$ for diameters above 10 mm. Measurements for at least 5 discrete diameter values have to be present in each event for it to be included in the comparison. Due to the scarcity of such events in the test set, the values of the error metrics for each of them are displayed individually as triangular markers.





**Table 1.** Error metrics from the comparison of $N_u(D)$ and $\hat{N}_u(D)$ for the drone event. The names and values of the metrics are shown in separate columns. Different parts of the HSND have been used for computing the metrics shown in the two rows of the table: the top row shows the metrics for the full HSND, while only diameters above 10 mm have been used to compute the ones in the bottom row.

|  | Bias [#/bin] | RMSE [#/bin] | Relative bias [%] | PCC [-] |
| --- | --- | --- | --- | --- |
| All diameters | -312 | 638 | -32.8 | 0.974 |
| $D > 10$ mm | -66.7 | 136 | -29.2 | 0.987 |



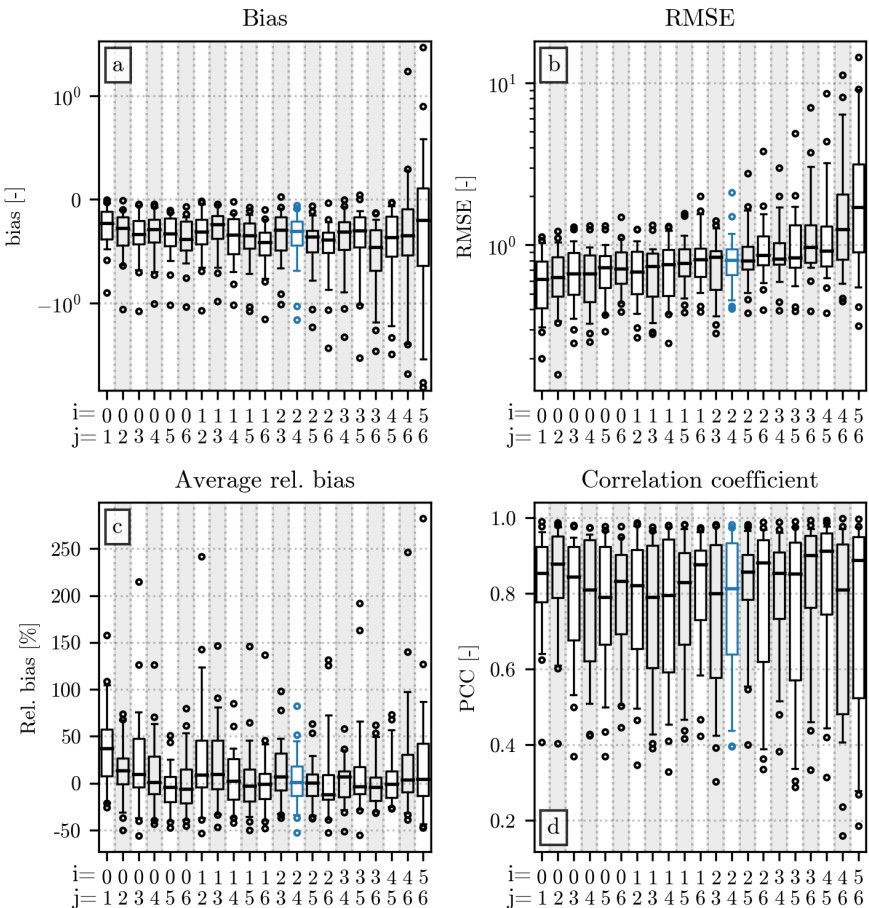

**Figure A1.** Comparison of the fitted $\hat{h}(x)$ with the normalized distributions in the test set, using all available pairs of moment orders, whose value is displayed on the x-axis. Each panel shows a different error metric, following the same order as Figure 3. The distribution of each metric is shown as a boxplot, with a structure analogous to the one used for the relative bias in Figure 3. The pair $[2, 4]$, used in the main body of the article, has been highlighted in blue. The scale of the y-axis in panel a is linear between -1 and +1, and logarithmic outside this interval.

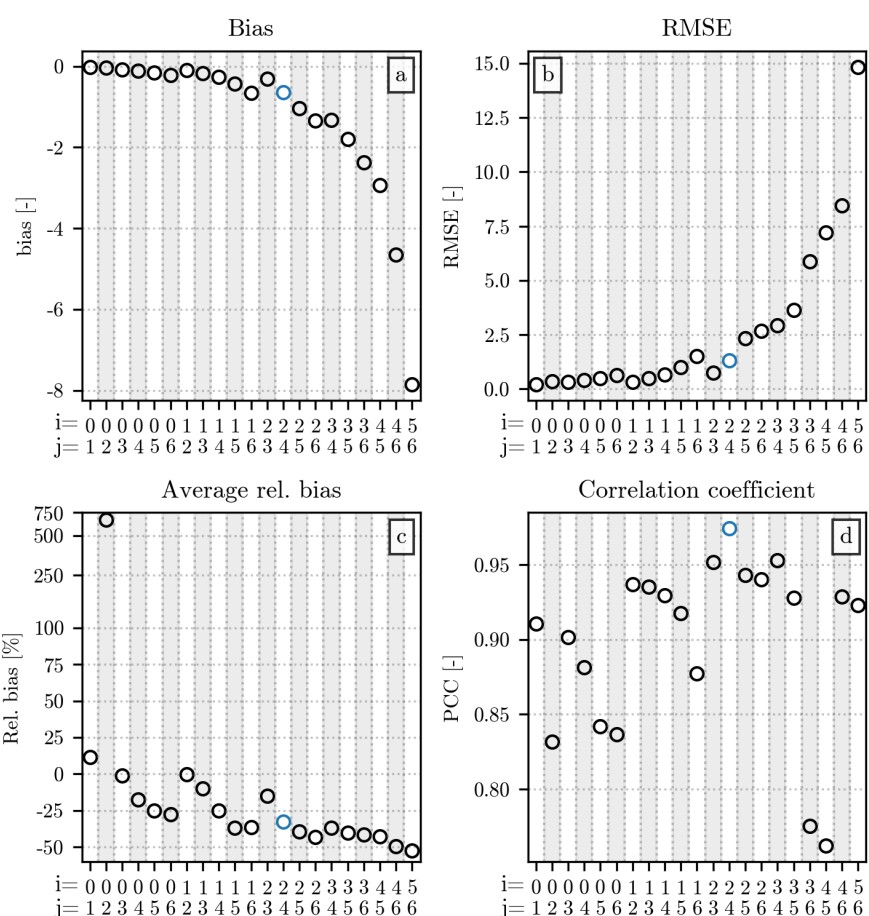

**Figure A2.** As Figure A1, but for the drone-derived dataset. Since the latter contains only one event, a single marker has been used instead of boxplots. The scale of the y-axis in panel d is linear between -100 and +100, and logarithmic outside this interval.



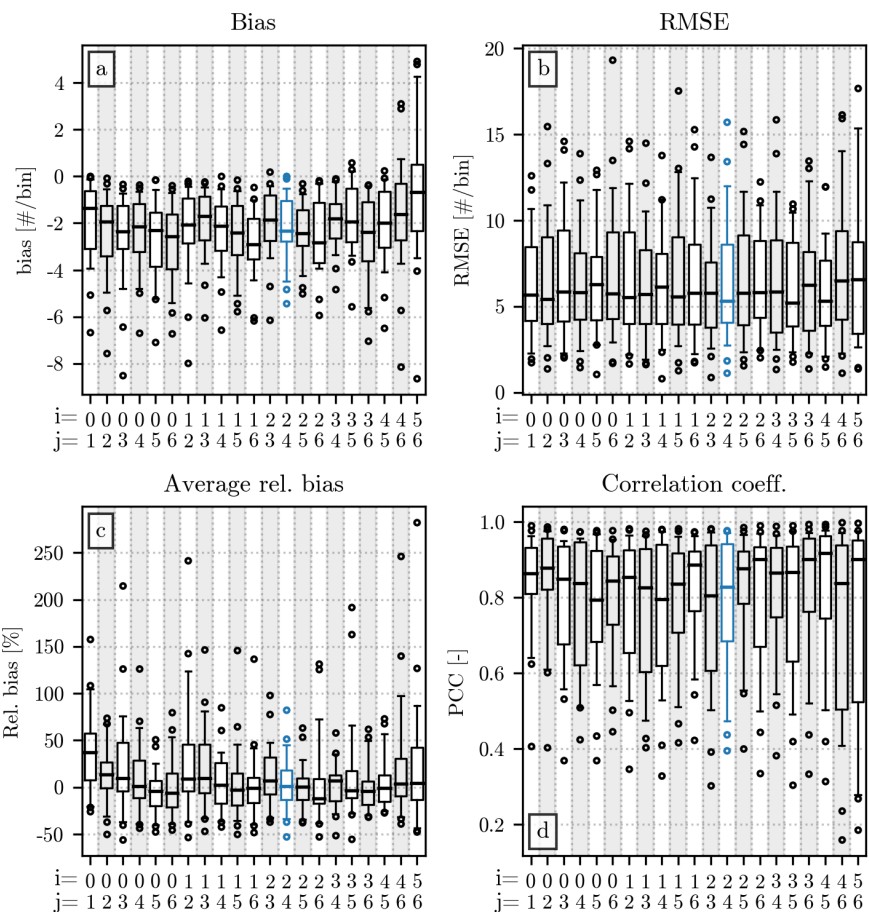

**Figure A3.** Comparison of $\hat{N}_u(D)$ with $N_u(D)$ for the events in the test set, using all available pairs of moment orders. The figure follows a structure analogous to the one of Figure A1. The extend of the y-axis in panel b has been limited between 0 #/bin and 20 #/bin to improve its readability, resulting in the exclusion of a single data point from the plot: for the moments pair $[5,6]$, one event has a value of RMSE close to 36 #/bin.



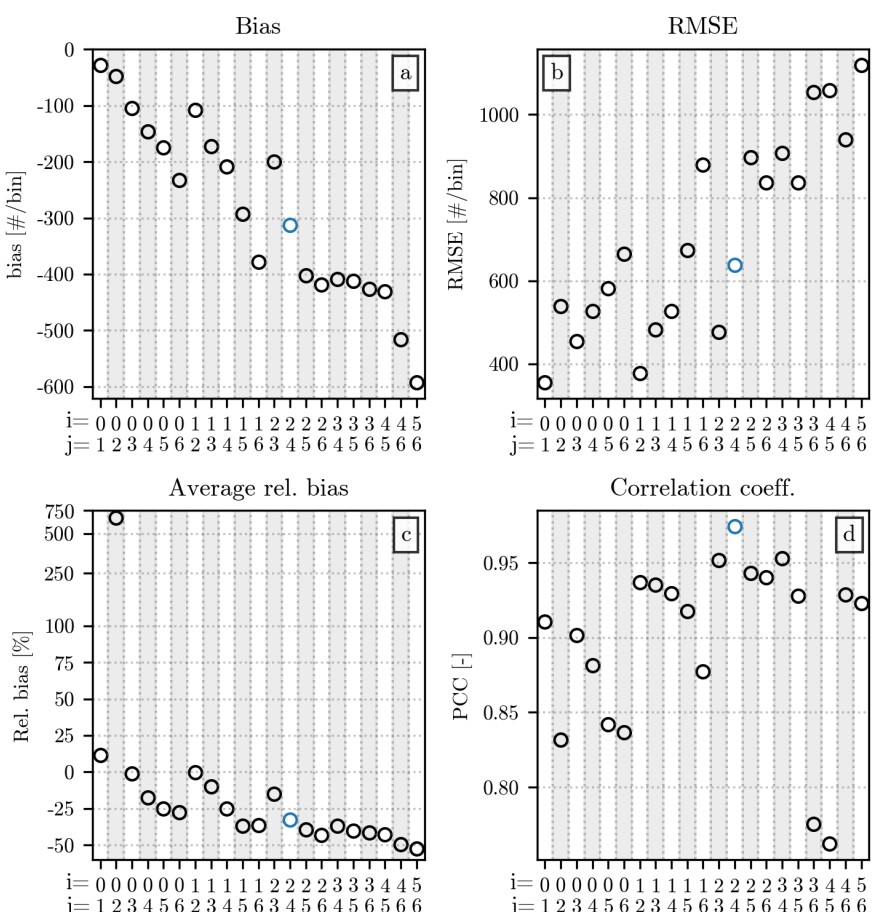

**Figure A4.** As Figure A2, but for the comparison of $\hat{N}_u(D)$ with $N_u(D)$, instead of the normalized distribution. The scale of the y-axis in panel d is the same as in panel A2.d.





**Table A1.** Parameters $c$ and $\mu$ of the generalized gamma model resulting from its fit over the training set, using different pairs of moment orders.

| Moment pair | $c$ [-] | $\mu$ [-] |
|---|---|---|
| 0, 1 | 0.11 | 500 |
| 0, 2 | 0.11 | 500 |
| 0, 3 | 0.12 | 500 |
| 0, 4 | 0.13 | 500 |
| 0, 5 | 0.14 | 500 |
| 0, 6 | 0.52 | 34 |
| 1, 2 | 0.11 | 500 |
| 1, 3 | 0.12 | 500 |
| 1, 4 | 0.12 | 500 |
| 1, 5 | 0.45 | 37 |
| 1, 6 | 1.1 | 5.2 |
| 2, 3 | 0.13 | 500 |
| 2, 4 | 0.37 | 46 |
| 2, 5 | 1.1 | 4.2 |
| 2, 6 | 1.6 | 1.8 |
| 3, 4 | 1.1 | 3.3 |
| 3, 5 | 1.6 | 1.3 |
| 3, 6 | 2.8 | 0.011 |
| 4, 5 | 2.8 | 0.000001 |
| 4, 6 | 2.7 | 0.000001 |
| 5, 6 | 2.7 | 0.000001 |