# Peer review of "Double moment normalization of hail size number distributions over Switzerland"

_Atmospheric Measurement Techniques, 2024_

## Referee Comment (RC1)

**Review of doi.org/10.5194/amt-2024-2**
**"Double moment normalization of hail size number distributions over Switzerland"**
by Alfonso Ferrone, Jérôme Kopp, Martin Lainer, Marco Gabella, Urs Germann, and Alexis Berne
Submitted to *Atmospheric Measurement Techniques*, April 2024.

**Summary**

In this work the authors apply a recently introduced –but for rain– method to describe the distribution of hailstone diameters, observed by the Swiss network of 80 automatic hail sensors, based on distribution "moments". In practice, for each of the 95 hail events studied (each one with a different time duration), they can compute the "$p$-moment", $M_p$, as the scalar quantity defined by:

$$M_p = \int_{5\,mm}^{D_{max}} D^p N(D) \, \mathrm{d}D \tag{1}$$

where $N(D)$ is the probability density function describing the distribution of hailstone diameters, $D$, of that individual event and $D_{max}$ is the maximum hailstone diameter observed by that specific hail event ($D_{max} \leq 20\,\mathrm{mm}$, a part for very few cases). Following what done in literature for rain, the authors rescale the hailstone size distribution introducing the "normalized diameter":

$$x = \left( \frac{M_i}{M_j} \right)^{\frac{1}{j-i}} \cdot D \tag{2}$$

and obtaining the "normalized distribution":

$$h(x) = \left( \frac{M_j^{i+1}}{M_i^{j+1}} \right)^{\frac{1}{j-i}} \cdot N(x(D)) \tag{3}$$

All the event distributions are converted in their normalized counterpart because $h(x)$ vs $x$ should "collapse" in a smaller area than $N(D)$ vs $D$ (even if it is not explained *why* that should happen). Thanks to this property, the authors are then able to fit a single gamma distribution $\hat{h}(x)$ on the space $h(x)$ vs $x$ to fit all the individual distributions with a single function. From this single function, they can derive the estimated $\hat{N}(D)$ distribution for each event and compare it with the observed one.

To verify their approach the authors divide all the 95 events in a training set (70% of cases used to fit $\hat{h}(x)$) and a test set (30%), using the distributions of BIAS, RMSE and R (Pearson correlation coefficient) as a measure of performance. Moreover, they use the same normalized distribution $\hat{h}(x)$ built from automatic hail sensors to fit a single hail event observed by drone on a large area. The last is characterized by a much higher number of hailstones and also larger diameters. The result is quite good, in particular as shape of the distribution, even if it tends underestimate $N(D)$. Lastly, since the hail sensor network is divided in three different subareas, the authors tried to compare the performance of one area relatively to the other two, finding that Ticino seems to be more

different from Jura and Napf area.

In conclusion, I found this work very interesting and promising, but I have a long list of suggestions and hence kindly ask for a relatively long revision, as suggested below.

**Major comments**

- One of the most annoying part of the paper is the somewhat excessive/not-friendly notation used. The authors introduce HSD, HSND, $N(D)$, $N_u(D)$ probably to indicate only two different concepts. The term "normalized" is probably used with different meaning: they say that HSD (hail size distribution, same as $N(D)$?) is *"usually computed over a unit area and for a fixed duration of time"*, hence one can think that it is "normalized" with respect to time and area, while the distribution studied here, HSND (hail size number distribution, same as $N_u(D)$?), is "unnormalized" because *"it is dependent on the detection area of the instrument and the duration of the event."*. Here there is already a problem, since all the hail sensors have the same area and hence only the single event observed by drone has a different detection area. But, a part from that, then the main topic of the work is to transform $N_u(D)$ into the normalized distribution $h(x)$ (eq. 3). In which sense reshaping $N_u(D)$ with equation 3 (where $N(x(D))$ is simply multiplied by the scalar value $\left(\frac{M_j^{i+1}}{M_i^{j+1}}\right)^{\frac{1}{j-i}}$) "normalize" the distribution? Is $h(x)$ normalized by area and time in some way?

- In the manuscript there is a lot of math, while I'm missing simple equation that can make the concepts easier to be understood by the readers. For example, is it correct that the total number of hailstones, $N_i$, of the $i$-hail event is given by:

$$N_i = \int_{5\,mm}^{D_{max}} N_i(D)\,\mathrm{d}D \ ? \tag{4}$$

If so, why such simple equation is never shown? BTW, if that is correct, it seems to me that $N_i(D)$ is more a probability *density* function that a number of hailstones, as written in the paper. Please clarify if $N_u(D)$ is a number distribution or better a density distribution, that gives "number" only when multiplied by $\mathrm{d}D$ (as written in line 52).
Moreover, at the end the paper focus only on moments of order 2 and 4, thus all the complex equations above reduces (if I'm correct) to:

$$x = \sqrt{\frac{M_2}{M_4}} \cdot D \tag{5}$$

and

$$h(x) = \sqrt{\frac{M_4^3}{M_2^5}} \cdot N(x(D)) \tag{6}$$

If these are the final equations used in this work, why they are never shown? Moreover, why the values of $M_2$ and $M_4$ are never discussed? For example, comparing Fig. 2 with Fig. 4 it seems that $x \cong D/20$. Please, could you show the distribution of $\sqrt{\frac{M_2}{M_4}}$ for your hail events? For example, I would be very interested in seeing a scatterplot of $\sqrt{\frac{M_2}{M_4}}$ versus $D_{max}$ of each event, to see if there is any systematic dependence of the double-moment normalization from the maximum diameter observed the original distribution. The same can be done for $\sqrt{\frac{M_4^3}{M_2^5}}$.

- Speaking about the verification of the spatial invariance, I understand that Section 5.4 should clarify well the point, but it is not very clear how the events used from one subarea are compared with the events in the other two regions: is that done fitting all the events of one area to build a new $\hat{h}(x)$, which is then used to reconstruct $\hat{N}(D)$ in the other two regions? Or only the training subsample in one region?
  A part from that and considering also the fact that Ticino seems to have different characteristics from the other two regions, I think that a more robust approach would be to redo all the main work using as training set all the events from two regions and as test set all the events from the third region (e.g. Ticino). In fact, using sensor locations from all three regions in the training set (chosen randomly) will give less robustness to the spatial invariance result, in my opinion.

- The verification if done using 4 different metrics, but since there is a value for each event, at the end only the verification metric distributions are shown. It may happen that these metrics are not independent each other and that an event having, for example, a better BIAS could have a worse correlation, while in other cases these metrics are more independent (see Section 5.3.1. of Manzato et al. 2016). There is a well known tool to show many of the most used regression verification metrics together in a single diagram, that is, the Taylor diagram (Taylor 2001, 2005). I invite the authors to consider to plot all the event verifications in a single Taylor diagram, instead of using four different diagrams showing distributions. If there are too many points, they could use a sort of "density plot" in the Taylor diagram space. Of course, mine is just a suggestion.

**Minor comments**

41-42 *have a surface of similar scale.*
Maybe similar order of magnitude?

44 *northern Italy*
northeastern Italy

55 equation $N(D)$
Please add the equation number and $[units]$.

68 *and period of time for which it is used.*
Here is a good place to explain something more on why the double moment normalization should work, i.e. why it should collapse the single event distriutions in a closer space.

91 *the quantity that we model is not normalized over a unit surface*
OK, but all sensors have the same dection area, so it is a constant. Ar you sure that you can't simplify the notation $N_u(D)$?

136-139 *Therefore,... HSNDs.*
Please, could you clarify better?

141-143 *Each of them has been defined as a period in which hailstones are recorded with a gap between consecutive impacts of less than 20 minutes, corresponding to the largest blank period studied in Kopp et al. (2023a).*
20 min seems to me a little too long period, in particular reading the recommendation of Kopp et al. (2023a), which states *"we suggest using a 10 to 15 min Tmb in further studies"* Do you have a specific justification to take the longest blank period studied?

157 *94 events*
In the Conclusions you write 95 (line 526). Please double check.

161 *minimum values,..., reaching 9 additional impacts*
Please explain better if you speak of hail events or hailstone impacts per single event or what else.

165 *the scarcity of measurements for the largest diameters (e.g. above 20 mm),*
Please write somewhere how many cases do you have above 20 mm and which is the maximum diameter used in this study. I suspect that you are missing the tail of large hailstones because of a too short database and that could undermine the general value of your $\hat{h}(x)$.

Equation 3
Written in this way, $h(x)$ does not seem a function of $x$.

Equation 4
At the end of the work, could you write an analytical version of eq 4 with the values that you have found, so that if some reader would like to apply to his dataset your findings it will be easy to be done? In general, it would be nice if this work will give some more "practical" results and a little less math.

Section 3.2
Consider to use the Taylor diagram if possible.

267 *PCC*
Please replace here and elsewhere PCC with R.

274-275 *the randomness of the impacts on the relatively small surface of the hail sensor*
Here you can quote Grieser and Hill (2019).

275 *has the potential to be noticeable in the upper tail*
Please reformulate.

317-319 *The main benefit of computing a normalized distribution is the reduced spread of the h(x) values at each x when compared to the considerable variability of Nu(D) at each D. This effect, sometimes referred to as the "collapse" of the normalized distributions, is what allows us to fit $\hat{h}(x)$, as described in section 3.1.*
This part can be moved and further elaborated in the Introduction.

Section 5.2
It would be possible to consider a Pareto distribution?

394-395 *The Pearson correlation coefficient is the only metric in which the fitted $\hat{h}(x)$ has better performances for the drone event than for most of the hail sensor ones.*
Any idea why? Is it possible because the drone hail event has a broader hail size distribution?

Section 5.4
Not clear which data exactly are used to train on one region and which to test on the other two.

458 *represents a middle ground*
Please reformulate.

474 *is linked with Nu(D) being an integer number*
Not yet clear if $N_u(D)$ is a number of hailstones or a density that must be multiplied by d$D$ to get an integer number.

572-574 *By finding a link between the empirical moments of the HSND and the radar measurements, it would be possible to use the formula and parameters of $\hat{h}(x)$ defined in this study to estimate the full distribution of hail diameters expected at the ground.*
That sounds very interesting. Could you elaborate further?

Appendix
It seems to me that there are not much differences (or using the words of the authors: *"Overall, the comparison between the various combinations of moments does not show any of the pairs clearly outperforming the others."*) thus the choice of moments 2 and 4 will never be very convincing. That is very evident already in Fig. A1, which I suggest to include in the main text.

Figures:
Plase make them larger. A page like that showing Fig. 2 is not acceptable.

Figure 3
The caption seems really too long.

Fig. 6 caption *The comparison is always performed between pairs of events,*
Not clear how the pair are chosen.

Fig. 9 colorbar
Please use more different colors, not only two.

**Suggested References**

I believe that this work could benefit considering at least part of the following references:

- Grieser, J. and Hill, M. (2019) How to express hail intensity modeling the hailstone size distribution. Journal of Applied Meteorology and Climatology, 58, 2329-2345. https://doi.org/10.1175/JAMC-D-18-0334.1.

- Manzato, A., Cicogna, A. and A. Pucillo. 2016. 6-hour maximum rain in Friuli Venezia Giulia: Climatology and ECMWF-based forecasts, Atmospheric Research, 169B, 465-484.

- Taylor, K.E. (2001). Summarizing multiple aspects of model performance in a single diagram. J. Geophys. Res. 106: 7183-7192. Bibcode:2001JGR...106.7183T. doi:10.1029/2000JD900719

- Taylor, K.E. Taylor diagram primer (2005), PDF available at https://pcmdi.llnl.gov/staff/taylor/CV/Taylor_diagram_primer.pdf.

Best Regards.

Tino

---

## Author Comment (AC1)

**Reply to RC1 and RC2**
**for the preprint "doi.org/10.5194/amt-2024-2"**

15 July 2024

We thank the reviewers for their valuable comments and suggestions which helped improve the quality of the manuscript in the revised version.

This document contains the replies to the two reviews, presented in separate sections. In the following, the comments from the reviewers are written in italics and highlighted in green. Our replies are written in standard black font, while direct quotations from the manuscript are shown in purple.

**1. Reply to RC1**

**1.1 Major comments**

*One of the most annoying part of the paper is the somewhat excessive/not-friendly notation used. The authors introduce HSD, HSND, N(D), Nu (D) probably to indicate only two different concepts. The term "normalized" is probably used with different meaning: they say that HSD (hail size distribution, same as N(D)?) is "usually computed over a unit area and for a fixed duration of time", hence one can think that it is "normalized" with respect to time and area, while the distribution studied here, HSND (hail size number distribution, same as Nu (D)?), is "unnormalized" because "it is dependent on the detection area of the instrument and the duration of the event.". Here there is already a problem, since all the hail sensors have the same area and hence only the single event observed by drone has a different detection area. But, a part from that, then the main topic of the work is to transform Nu (D) into the normalized distribution h(x) (eq. 3). In which sense reshaping Nu (D) with equation 3 (where N(x(D)) is simply multiplied by the scalar value*

$$\left( \frac{M_j^{i+1}}{M_i^{j+1}} \right)^{\frac{1}{j-1}}$$

*"normalize" the distribution? Is h(x) normalized by area and time in some way?*

Thank you for pointing out the issue in the nomenclature. Indeed, the term "normalized" indicated two different things in the preprint, which was a potential source of confusion.

In the revised manuscript we use "normalized" only when referring to quantities related to the double-moment normalization (e.g. the normalized distribution h(x) ), but not in the introduction when referring to N(D).

Regarding the choice of using $N_u(D)$ (and therefore, without dividing the number of impacts by the detection area), we discuss it in detail in the answer to the minor comment for line 91 of the preprint.

Finally, we use the term "normalized" when referring to h(x) for consistency with existing literature, both for the study of DSD (Lee et al, 2004, Raupach and Berne 2017) and for the study of hail (Field et al, 2019). While h(x) is not normalized by area or time, its value is independent of the value of the moment (whose value depends on the area, as well as other factors such as the event's intensity). The scaling factor:

$$\left( \frac{M_j^{i+1}}{M_i^{j+1}} \right)^{\frac{1}{j-1}}$$

is what allows the conversion of h(x) to the measured distribution (basically "adding back" the information on the value of the moment). A good explanation of this procedure is provided in the introduction of the paper by Lee et al. (2004).

In the revised manuscript, we clarified this point by slightly rephrasing how the normalized distribution is introduced. The new version is the following:

To address some issues with different rainfall types, Lee et al. (2004) introduced the double-moment normalization, which allows representing a DSD using two of its moments and the knowledge of the "normalized distribution". The latter provides the overall "shape" of the DSD and is assumed to be invariant over the geographical region and period of time for which it is used. This property of the normalized distribution is linked to its independence from the value of the moments of the distribution, as explained by Lee et al (2004). When compared to the method proposed by Sempere-Torres (1994), the double moment normalization better captures the natural variability in the shape of the distributions, thanks to the addition of the second moment.

*In the manuscript there is a lot of math, while I'm missing simple equation that can make the concepts easier to be understood by the readers. For example, is it correct that the total number of hailstones, Ni , of the i-hail event is given by:*

$$N_i = \int_{5mm}^{D_{max}} N_i(D)\, dD$$

*If so, why such simple equation is never shown? BTW, if that is correct, it seems to me that Ni (D) is more a probability density function that a number of hailstones, as written in the paper. Please clarify if Nu (D) is a number distribution or better a density distribution, that gives "number" only when multiplied by dD (as written in line 52).*
*Moreover, at the end the paper focus only on moments of order 2 and 4, thus all the complex equations above reduces (if I'm correct) to*

$$x = \sqrt{\left(\frac{M_2}{M_4}\right)}\, D$$

*and*

$$h(x) = \sqrt{\left(\frac{M_4^3}{M_2^5}\right)}\, N(x(D))$$

*If these are the final equations used in this work, why they are never shown? Moreover, why the values of M2 and M4 are never discussed? For example, comparing Fig. 2 with Fig. 4 it seems that x = D/20. Please, could you show the distribution of √(M₂/M₄) for your hail events? For example, I would be very interested in seeing a scatterplot of √(M₂/M₄) versus D_{max} of each event, to see if there is any systematic dependence of the double-moment normalization from the maximum diameter observed the original distribution. The same can be done for √(M₄³ / M₂⁵).*

We agree that the inclusion of the proposed equations would benefit the clarity of the manuscript. The reviewer is correct in their formulation of the total number of hailstones, which has been included as equation 3 (section 3.1) of the revised manuscript.

$N_u(D)$ is not a probability density function, since its integral is not 1. But the reviewer is correct in pointing out that ultimately it is a density. The units in the figures already reflected this nature (#/bin), but the information was not explicitly presented in the text. This lack of information has been addressed in the revised manuscript, in which $N_u(D)$ is now described by:

Which such a binning in place, $N_u(D)$ represents a density of hailstone per diameter bin, and it is expressed in number of impacts (#) per bin.

In the same section, we also updated the formula for the moments to include the correct lower limit of the integral (5mm).

The formulation of x and h(x) for the final choice of moments has also been added to the revised manuscript in section 4. The reviewer is also correct in pointing out that the moments values were not discussed in the manuscript. We decided to include an additional figure in Appendix A to show these values:

[Figure]

We think that it could be useful in providing additional information when analyzing the error metrics presented in Figure 3.

Following the suggestion of the reviewer, we briefly investigated the relationship between the moments and the value of D_max, even though we did not include this additional analysis in the manuscript.

We decided first to have a lootk at the scatterplot between moment values and $D_{max}$ for each event (figure on the right). The higher the moments, the closer the points are to the diagonal of each panel. This behavior is expected since the largest diameter has a higher impact on the value of the moment when the order is higher.

On the next page, we provide two additional figures:
- the first shows the histogram of the value of $\sqrt{(M_2/M_4)}$ and $\sqrt{(M_4^3 / M_2^5)}$ in the left and central panels, and the scatterplot between the two quantities in the panel on the right;
- the second one shows the scatterplots of $\sqrt{(M_2/M_4)}$ and $\sqrt{(M_4^3 / M_2^5)}$ versus the maximum diameter.

In both figures, each marker represents a hail event.

[Figure]

[Figure]

The first figure reveals that higher values of $\sqrt{(M_4^3 / M_2^5)}$ generally correspond to smaller values of $\sqrt{(M_2/M_4)}$, but the points do not lie on a clear line, and instead they are spread out considerably.
In the second figure, instead, there is a clear relationship between $\sqrt{(M_2/M_4)}$ and $D_{max}$. As seen in the figure on the previous page, there is a relationship between individual moments and the maximum diameter, which becomes stronger for high moment orders. This relationship is probably responsible for the one we see in the second figure of this page.
The relationship between $\sqrt{(M_4^3 / M_2^5)}$ and $D_{max}$ is, instead, less clear (we attempted to plot it with a log scale on the y-axis, but this change did not make any relationship more visible).

Overall, the existence of relationships between the value of the moments and $D_{max}$, or the one between the scaling factor $\sqrt{(M_2/M_4)}$ and $D_{max}$, is to be expected. The independence of h(x) from the value of the moments is what allows it to be "universal" (at least over the locations used in the study). This "universality", in turn, allows us to fit h(x) with a generalized gamma. Then, the scaling factor allows us to use the fitted normalized distribution for computing an estimate of $N_u(D)$, and therefore it must reintroduce information on the moments, which in turn depends on the value of the hailstone diameters (and its dependence from the largest diameter increases with the moment order).

Finally, the formulas for x and h(x) for moments 2 and 4 have been added to the revised manuscript.

Speaking about the verification of the spatial invariance, I understand that Section 5.4 should clarify well the point, but it is not very clear how the events used from one subarea are compared with the events in the other two regions: is that done fitting all the events of one area to build a new ĥ(x), which is then used to reconstruct N̂(D) in the other two regions? Or only the training subsample in one region?
A part from that and considering also the fact that Ticino seems to have different characteristics from the other two regions, I think that a more robust approach would be to redo all the main work using as training set all the events from two regions and as test set all the events from the third region (e.g. Ticino). In fact, using sensor locations from all three regions in the training set (chosen randomly) will give less robustness to the spatial invariance result, in my opinion.

In the revised manuscript, we added a short clarification in section 5.1, in which the training and test sets have been defined.
The two sets are introduced by the following text in the revised article:
The training and test sets contain approximately 70% (67 events) and 30% (27) of the whole hail sensor dataset, respectively. Data are assigned to the two sets using the Python pseudo-random number generator (Python Software Foundation, 2023)  and following a simple set of rules: no hail sensor location can appear in both sets, and the data from the three regions (Jura, Napf, Ticino) must follow the same 70%-30% split. Therefore, data from each region appear in both the training and test sets. Results from an alternative set definition, in which regions are kept separated, are presented in section 5.4.

Using this approach, the fit of the normalized distribution uses information from all the regions and it will (theoretically) be better suited for practical applications over the whole Swiss territory. However, we agree with the reviewer on the need to also test the fit over the three regions separately, to better confirm the "universality" of h(x).
Therefore, we decided to repeat the fit three times, using each time one of the three possible pairs of regions for the training, and the remaining one for the test set.
The error metrics that compare the fitted normalized distribution with the one measured in each of the test sets have been included as an additional figure in the revised manuscript:

[Figure]

Interestingly, Ticino (used as test set) does not stand out for its differences when compared to the other two regions. Higher values of RMSE and lower values of correlation can be seen for Napf (but they remain somehow similar). More noticeably, a higher spread around the median values can be seen for Napf. This may be due to the larger amount of hail events recorded in this region, which could result in a larger variety of events in the test set and a smaller pool of data in the training set that can be used for the fit.

Overall, the value of the metrics remains close to the ones displayed in Figure 5 of the preprint, which have been computed using a normalized distribution fitted over events from all three regions. In our opinion, this result suggests that even though some differences may exist between the three regions, the assumption of invariance between the three regions is reasonable.

The verification if done using 4 different metrics, but since there is a value for each event, at the end only the verification metric distributions are shown. It may happen that these metrics are not independent each other and that an event having, for example, a better BIAS could have a worse correlation, while in other cases these metrics are more independent (see Section 5.3.1. of Manzato et al. 2016). There is a well known tool to show many of the most used regression verification metrics together in a single diagram, that is, the Taylor diagram (Taylor 2001, 2005). I invite the authors to consider to plot all the event verifications in a single Taylor diagram, instead of using four different diagrams showing distributions. If there are too many points, they could use a sort of "density plot" in the Taylor diagram space. Of course, mine is just a suggestion.

We thank the reviewer for the suggestion. We decided to attempt to plot the error metrics (just for the HNSD reconstruction) in a Taylor diagram, and the result is the following:

[Figure]

While the diagram is a useful tool, we prefer the readability of how the error metrics have been presented in the preprint (e.g. immediate visualization of the median value of each metric).

Regarding the relationship between error metrics, we decided to include on the next page a scatterplot of the error metrics computed for the HSND reconstruction (test set), using the moment pair 2-4.

[Figure]

A negative correlation between bias and RMSE exists, while the relative bias and the RMSE are positively correlated (with a considerable spread). The relationship between other error metrics is less clear.

In our opinion, RMSE and bias offer valuable and different information, despite their correlation. The bias clearly shows how the fitted h(x) and the reconstructed HSND underestimate their counterpart. Since this underestimation is not constant across the interval of x and D values, the RMSE gives us a better estimate of the difference in the number of impacts (or value of h(x)) per bin. The relative bias, too, provides information that differs from the previous metric. For example, a difference of 10 impacts between the measured and reconstructed HSNDs will be more or less relevant depending on the number of hailstones in the specific diameter bin where this difference has been recorded. If this difference is recorded for small diameter values, for which tens of impacts have been recorded, the overall effect on the distribution is smaller, in relative terms, than the one resulting from the same difference being recorded at high D values, for which only a handful of hailstones are observed.

For these reasons, and since the same error metrics have been used in previous literature when applying the double-moment normalization (Raupach and Berne, 2017), we prefer to keep them in the revised manuscript.

**1.2 Minor comments**

*41-42 have a surface of similar scale.*
*Maybe similar order of magnitude?*
Thank you for the correction, we included it in the revised version of the manuscript.

*44 northern Italy*
*northeastern Italy*
The correction has been included in the revised manuscript.

*55 equation N(D)*
The equation number has been added and the units ($m^{-3}cm^{-1}$) have been specified in the revised manuscript.

*68 and period of time for which it is used.*
*Here is a good place to explain something more on why the double moment normalization should work, i.e. why it should collapse the single event distriutions in a closer space.*
We added a short sentence in the revised manuscript to explain how this property arises from the independence of h(x) from the moments, also citing again the work of Lee et al. (2004), which provides the underlying theory.

*91 the quantity that we model is not normalized over a unit surface*
*OK, but all sensors have the same dection area, so it is a constant. Ar you sure that you can't simplify the notation Nu (D)?*
While a simplification would be possible, we prefer to keep the current notation.
Dividing the number of impacts by the area would just add a constant factor for all events recorded by the hail sensors, and therefore would not alter the shape of the distributions used for the fit. Furthermore, even though the drone event is just one, dividing its number of recorded impacts by the area would add an unnecessary source of uncertainty since we would have to estimate the exact area in which the drone was able to effectively record hailstones. While it would be possible to produce such an estimate, the same may not apply to future users of the method. A hypothetical future user may not have access to the area of detection of hailstones, resulting in an unnecessary limitation to the applicability of the method. The approach selected now, instead, can be used even without this knowledge.

*136-139 Therefore,. . . HSNDs.*
*Please, could you clarify better?*
We rephrased it as:
Excluding all measurements below these thresholds would, however, greatly reduce the number of data available for the analysis, undermining its robustness. For example, the fit of the generalized gamma function over the normalized distributions, described in section 5.2, relies on the availability of a dataset as large as possible. The quality of this fit, and of the results deriving from it, would decrease if we were to remove all data from the range of diameters in which the highest number of impacts has been recorded.

*141-143 Each of them has been defined as a period in which hailstones are recorded with a gap between consecutive impacts of less than 20 minutes, corresponding to the largest blank period studied in Kopp et al. (2023a).*
*20 min seems to me a little too long period, in particular reading the recommendation of Kopp et al. (2023a), which states "we suggest using a 10 to 15 min Tmb in further studies" Do you have a specific justification to take the longest blank period studied?*

Thank you for the correction. The threshold was decided in an earlier stage of the manuscript, and we decided to use the largest time interval used by Kopp et al (2023a). We did not update it when the paper by Kopp et al. was published. In this new iteration, the threshold has been reduced to 15 minutes. The figures in the revised manuscript have been remade using this new dataset. The change in the results is marginal.

*157 94 events*
*In the Conclusions you write 95 (line 526). Please double check.*
Thank you for noticing the discrepancy in the text, we corrected it. The value in the conclusions was wrong, and now it has been replaced with the correct one (94).

*161 minimum values,. . . , reaching 9 additional impacts*
*Please explain better if you speak of hail events or hailstone impacts per single event or what else.*
We added the clarification "across all events" in the revised manuscript so that these 9 impacts can be correctly interpreted as a total number of impacts removed (and not a value per event).

*165 the scarcity of measurements for the largest diameters (e.g. above 20 mm),*
*Please write somewhere how many cases do you have above 20 mm and which is the maximum diameter used in this study. I suspect that you are missing the tail of large hailstones because of a too short database and that could undermine the general value of your ĥ(x).*
We added the following text in the revised manuscript to clarify the number of hailstones above 20 mm and the maximum diameter recorded:
In total, only 8 events in our dataset have hailstones with D > 20 mm, while the largest diameter recorded is D = 41.8 mm.

We are aware that the dataset is relatively small. However, the data we used are the only ones of this type available at the moment in Switzerland. Future studies, when more data have been collected, may compute the fitted normalized distribution again and provide more robust estimates. However, at the moment, even with a limited dataset, the proposed fit closely follows h(x) from the events (Figure 4), and the HSND reconstructed for the drone event (Figure 8) is similar to the original one, despite the considerable difference in number of impacts between hail sensors and drone. In our opinion, while further work is needed to determine h(x) with a high level of accuracy, the results presented in our study demonstrate the relevance of the double-moment normalization and can still be valuable in modeling the HSND.

Equation 3
Written in this way, h(x) does not seem a function of x.
Thank you for noticing this mistake. We have expanded the formula, providing also a version with an explicit dependence from x.

Equation 4
At the end of the work, could you write an analytical version of eq 4 with the values that you have found, so that if some reader would like to apply to his dataset your findings it will be easy to be done? In general, it would be nice if this work will give some more "practical" results and a little less math.
We added the versions of equations 4 and 5 for the choice of the moment pair 2-4 in section 4 (moments selection).

*Section 3.2*
*Consider to use the Taylor diagram if possible.*
A reply to this point has been included in the answer to the last of the major points of the review.

*Please replace here and elsewhere PCC with R.*

PCC has been replaced with R in the revised manuscript (in the text and in the figures).

*274-275 the randomness of the impacts on the relatively small surface of the hail sensor*
*Here you can quote Grieser and Hill (2019).*

Thank you for the suggestion, the citation has been added to the revised manuscript.

*275 has the potential to be noticeable in the upper tail*
*Please reformulate.*

The sentence has been reformulated in the revised manuscript as:

Furthermore, due to the scarcity of big hailstones, the randomness of the impacts on the relatively small surface of the hail sensor has the potential to affect the representativeness of the tail of the HSND.

*317-319 The main benefit of computing a normalized distribution is the reduced spread of the h(x) values at each x when compared to the considerable variability of Nu(D) at each D. This effect, sometimes referred to as the "collapse" of the normalized distributions, is what allows us to fit ĥ(x), as described in section 3.1.*
*This part can be moved and further elaborated in the Introduction.*

We further elaborated this part in the revised manuscript by adding an explicit mention of the independence of h(x) from the moment values, and how this is linked to its "universality".
In the introduction, as mentioned in a previous reply, we also added similar information.

However, we decided not to move this whole sentence to the introduction as suggested by the reviewer. The sentence relies on quantities, such as x and ĥ(x), which have not been introduced when the double-moment normalization is first explained. We prefer to keep their explanation in the section dedicated to the theory behind the method (section 3), where these quantities are defined using equations 4 and 6.

*Section 5.2*
*It would be possible to consider a Pareto distribution?*

In theory, all the distributions that satisfy the conditions detailed in the paper by Lee et al (2004) can be used for the ĥ(x) fit.
However, given the limited size of our dataset, we decided to use distributions of which we can find examples in previous literature on the double-moment normalization, both for DSD (e.g. Raupach and Berne, 2017) and hail (Field et al, 2019).
Since the hail sensor network is still recording hail events, it may be possible in future studies to refine the ĥ(x) fit, using a less conventional distribution.

*394-395 The Pearson correlation coefficient is the only metric in which the fitted ĥ(x) has better performances for the drone event than for most of the hail sensor ones.*
*Any idea why? Is it possible because the drone hail event has a broader hail size distribution?*

We are not sure of the reason behind it, but it is possible that the higher number of counts in the drone HSND, especially for higher diameter values, reduces the amounts of fluctuations in h(x), and therefore it aligns better with ĥ(x). In the revised manuscript, we added a short mention of this possibility.

*Section 5.4*
*Not clear which data exactly are used to train on one region and which to test on the other two.*

Information on how we addressed this point has been provided in our answer to the third major comment.

*458 represents a middle ground*
*Please reformulate.*
We rephrased it in the revised manuscript as:
The event in panel 8.c has a number of impacts in between the previous two.

*474 is linked with Nu(D) being an integer number*
*Not yet clear if Nu (D) is a number of hailstones or a density that must be multiplied by dD to get an integer number.*
In addition to the clarification on the nature of $N_u(D)$ mentioned previously in the reply, in this particular sentence we added the following text after the mention of $N_u(D)$ being an integer number:
since it represents the number of impacts recorded by the hail sensors in a specific diameter bin

*572-574 By finding a link between the empirical moments of the HSND and the radar measurements, it would be possible to use the formula and parameters of ĥ(x) defined in this study to estimate the full distribution of hail diameters expected at the ground.*
*That sounds very interesting. Could you elaborate further?*
Thank you for your interest in this topic. Indeed, this application is among the main reasons behind the submission of the current article. This study represents our attempt to investigate the possibility of describing the distribution of diameter sizes using well-defined quantities (e.g. the empirical moments). Other colleagues at EPFL and MeteoSwiss are now working on using the results of this study to examine whether a link exists between the HSND moments and various radar products. Similar studies have been conducted in the case of rain and drizzle.
In the ERAD 2024 meeting, Matteo Guidicelli will present the latest results on this potential application. In the current preprint, we decided not to elaborate further on this topic, since not all of its components have yet been completely defined and tested. However, in the following months, an article on this application will likely be submitted by an EPFL/MeteoSwiss team.

*Appendix*
*It seems to me that there are not much differences (or using the words of the authors: "Overall, the comparison between the various combinations of moments does not show any of the pairs clearly outperforming the others.") thus the choice of moments 2 and 4 will never be very convincing. That is very evident already in Fig. A1, which I suggest to include in the main text.*
We agree with the reviewers that the small difference in performances between the moment pair 2-4 and other pairs should be better explained in the main text of the article. We included additional information at the end of section 4 in the revised manuscript.
However, we decided not to include Figure A1, since its explanation would require moving the content from Appendix A into the main text. The content of the appendix has been presented separately from the main text since it is not crucial to the understanding of the method, while still being potentially useful for a reader interested in applying it.

*Figures:*
*Plase make them larger. A page like that showing Fig. 2 is not acceptable*
We followed the guidelines on the size of the figures indicated in the template provided by Copernicus.
Single-panel figures must have a width of 8.3cm, which is the case for Figure 2. Even if we were to include a larger figure in the preprint, it would be resized to a width of 8.3 cm in the final version of the manuscript.
It would be possible to increase its size by making it span 2 columns, which would allow us to make it 12 cm wide. However, we do not think that all figures in a paper must span the whole width of the page. In the case of Figure 2, it consists of a single panel, wider than the panels in the following figures that occupy 2 columns, and therefore we think it is appropriate for it to be placed as a single-column figure in the text.

*Figure 3*
*The caption seems really too long.*

We slightly reduced the length of the caption in the revised manuscript.

*Fig. 6 caption The comparison is always performed between pairs of events,*
*Not clear how the pair are chosen.*

We specified in the caption in the revised manuscript that the comparison is performed between all possible combinations of pairs of events.

*Fig. 9 colorbar*
*Please use more different colors, not only two.*

We changed the colorbar, selecting one with four colors: black, red, yellow, and white.

**1.3 Suggested References**

I believe that this work could benefit considering at least part of the following references:

- Grieser, J. and Hill, M. (2019) How to express hail intensity modeling the hailstone size distribution. Journal of Applied Meteorology and Climatology, 58, 2329-2345. https://doi.org/10.1175/JAMCD-18-0334.1.
- Manzato, A., Cicogna, A. and A. Pucillo. 2016. 6-hour maximum rain in Friuli Venezia Giulia: Climatology and ECMWF-based forecasts, Atmospheric Research, 169B, 465-484.
- Taylor, K.E. (2001). Summarizing multiple aspects of model performance in a single diagram. J. Geophys. Res. 106: 7183-7192. Bibcode:2001JGR...106.7183T. doi:10.1029/2000JD900719
- Taylor, K.E. Taylor diagram primer (2005), PDF available at https://pcmdi.llnl.gov/staff/taylor/CV/Taylor diagram primer.pdf.

We have included the reference to the paper of Grieser and Hill (2019). Since the other suggested references have been proposed in the context of using the Taylor diagram as error metric, they have not been included in the revised manuscript.

**2. Reply to RC2**

*I think that the general approach of double moment normalisation, the segregation of the data into training and test datasets and the quantification of bias and root mean square error is all good. I am particularly excited by the use of drone observations. This drone imagery approach could vastly improve the statistics of hail observations from the ground and the comparison to traditional hail pads supports this. Perhaps more could be made of this in the conclusions?*

Thank you, we also share your enthusiasm for the potential of drone observations in this field. Many practical aspects limit the number of events that can be captured by this technology, but when observations are available, the number of hailstones recorded is considerably larger than the one typically seen by other means, such as the hail sensors.

In our opinion, drone observations at the moment represent the ideal source of data for verification (thanks to the "completeness" of the distribution), but it would be difficult to use the same data to propose the use of a new method. For instance, we could not justify the usage of one (or a few) drone events for fitting a normalized distribution that should be "universal" over Switzerland. However, if more campaigns are conducted in the future, a larger dataset could indeed be used for updating the fit, hoping to better capture the shape of the distribution at high diameter values.

In the revised manuscript, we slightly expanded the conclusions after the sentence:
*Additional measurement campaigns based on drone imagery may also provide sets of events with a high number of impacts for a correct evaluation of the reconstructed HSND for relatively large hailstone diameters.*

We added the following text to highlight the potential for additional drone applications:
*If a high enough number of drone observations are collected across different regions of Switzerland, it may also be possible to use them for further refining the fit of h(x), thanks to the larger amount of data available for each x bin.*

*The data is all fine - i think that all that is needed is to revisit the fitting again, perhaps taking into account the missing or truncated data. Maybe consider different models and it would be good to look at emphasizing results for moment choices that would allow this work to be used more easily by modelers (e.g. moments, 0,3) and people doing retrievals (e.g. radar:6). I think this would be major revision.*
When we first decided to apply the double-moment normalization for modeling the HSND, we also thought of the usefulness of choosing moments 3 and 6 for the study, especially for a potential radar retrieval in the future. However, the results shown in Appendix A, together with the sensitivity of the value of moment 6 to missing hailstones at the tail of the distribution, led us to a different choice of moments.

Nevertheless, as the reviewer points out, it would still be useful to give more emphasis to moment pairs such as 3-6, despite their poor performances shown in Appendix A. To do so, we decided to include in the manuscript a second appendix, "Appendix B".
In this second appendix, we propose a different (and simpler fit) of h(x): an exponential. Due to the restriction listed by Lee et al (2004), the only exponential curve suited for fitting h(x) is the one obtained by substituting the two parameters of the generalized gamma with a value of 1.
This approach works relatively well only for high moment values, which are the focus of this second Appendix.
In the main text, however, we decided to keep moments 2-4 as the selected pair, since their performances are (slightly) better than the ones of other pairs.

**2.1 Main points**

*1. The choice of moments. Moments 2 and 4 are difficult to relate to physical quantities. Perhaps number (0), mass (~3), radar reflectivity (~6) would be more useful for others to use for radar retrievals (as mentioned in the conclusion) or in models that predict mass and potentially number. The fits are given for those pairs of moments but they look to have not converged.*
Thank you for the suggestion. This topic has been covered in our reply to the previous point of this review.

*2. The fit values for c, mu. The fits are done in log space using the normalized bin value and bin centers so that data with widely varying sample times can be used (i think). What is done for the bins where N(x)dx =0? Are these bins ignored for the fitting? Or is some large negative value in log space assumed? Some of the fit values for the exponent on x are large (c=0.37, mu=46 for moments 2,4, giving an exponent of 16). Are these large exponents a function of trying to fit the unobserved values at the small end that are assigned a small value? Should a simpler model for the size distribution be used instead that will converge for all moment choices?*
While the logarithmic scale has been used for the y-axis in some figures (otherwise values at high x would be unreadable), the fit has been done in linear units. In the revised manuscript we added a clarification on this topic in section 5.2.

The values of c and mu are indeed high, as noted by the reviewer. As highlighted in the manuscript, the generalized gamma used for the fit struggles for small diameter values, and we think that the high c or mu values are caused by the difficulty of the chosen function to capture the shape of the distribution in this region.
We tried to use a simpler model (exponential), but the value of the error metrics were worse than the ones obtained through the generalized gamma fit for many moment pairs. The only exceptions are moments with a high order. We decided to include the results for the exponential fit for these moments in Appendix B in the revised manuscript.

*3. The c and mu values that have not converged (i.e. where the fit values are at one end (e.g. 500) or the other of the allowed range) should probably not be considered useful and there will be no need to report on their bias and rmse. I'm slightly worried that some non-converged fits (e.g. 0,1, 0,5) have less bias and rmse than those that have (e.g. 2,4). The very large mu values are unconstrained because of the lack of observations. Again maybe a different model would be better?*
Thank you for the suggestion. In the revised manuscript, the figures of Appendix A have been remade, differentiating moment pairs for which the fit converged from the ones for which it did not.

We do not know for certain the reason why some moment pairs for which the fit did not converge have, for some metrics, better performances than other moment pairs for which the fit converged. It should be noted, however, that even in the pairs for which the fit did not converge, the chosen values of mu and c are the ones that minimize the RMSE between h(x) and ĥ(x) in the training set. The fit is probably trying to push the generalized gamma towards a shape that requires extremely high or low values for one of its parameters, and it is therefore limited by the boundaries imposed (to avoid numerical errors). This behavior may indeed suggest that the generalized gamma is not the function best suited to represent h(x) for these moment pairs, and this is one of the reasons why in the revised manuscript we decided to include Appendix B with a simpler shape for ĥ(x).
However, for some moment pairs (among which there is the one chosen for the main text,  2-4) the fit converges. We think that the chosen generalized gamma model is an acceptable shape for ĥ(x) in these cases since it produces better values for the error metrics than the one obtained by the exponential fit used in Appendix B.

**3. References**

The following references have been cited in the replies to RC1 and RC2:

Field, P. R., Heymsfield, A. J., Detwiler, A. G., and Wilkinson, J. M.: Normalized Hail Particle Size Distributions from the T-28 Storm-Penetrating Aircraft, Journal of Applied Meteorology and Climatology, 58, 231 – 245, https://doi.org/10.1175/JAMC-D-18-0118.1, 2019.

Kopp, J., Manzato, A., Hering, A., Germann, U., and Martius, O.: How observations from automatic hail sensors in Switzerland shed light on local hailfall duration and compare with hailpad measurements, Atmospheric Measurement Techniques, 16, 3487–3503, https://doi.org/10.5194/amt-16-3487-2023, 2023a

Lee, G. W., Zawadzki, I., Szyrmer, W., Sempere-Torres, D., and Uijlenhoet, R.: A General Approach to Double Moment Normalization of Drop Size Distributions, Journal of Applied Meteorology, 43, 264 – 281, https://doi.org/10.1175/1520-0450(2004)043<0264:AGATDN>2.0.CO;2, 2004

Raupach, T. H. and Berne, A.: Retrieval of the raindrop size distribution from polarimetric radar data using double moment normalisation, Atmospheric Measurement Techniques, 10, 2573–2594, https://doi.org/10.5194/amt-10-2573-2017, 2017.

Sempere-Torres, D., Porrà, J. M., and Creutin, J.-D.: A General Formulation for Raindrop Size Distribution, Journal of Applied Meteorology and Climatology, 33, 1494 – 1502, https://doi.org/10.1175/1520-0450(1994)033<1494:AGFFRS>2.0.CO;2, 1994.